# MIME: MIMICKING CENTRALIZED STOCHASTIC ALGORITHMS IN FEDERATED LEARNING

## ABSTRACT

Federated learning (FL) is a challenging setting for optimization due to the heterogeneity of the data across different clients. This heterogeneity has been shown to cause a *client drift*, which can significantly degrade the performance of algorithms designed for the FL setting. In contrast, centralized learning with centrally collected data is not affected by such a drift and has seen great empirical and theoretical progress with innovations such as momentum and adaptivity. In this work, we propose a general algorithmic framework, MIME, which mitigates client drift and adapts arbitrary centralized optimization algorithms such as SGD and Adam to the federated learning setting. MIME uses a combination of *control-variates* and *server-level statistics* (e.g. momentum) at every client-update step to ensure that each local update mimics that of the centralized method run on iid data. Our thorough theoretical and empirical analyses strongly establish MIME's superiority over other baselines.

## 1 INTRODUCTION

Federated learning has become an important paradigm in large-scale machine learning where the training data remains distributed over a large number of clients, which may be mobile phones or network sensors (Konečný et al., 2016b;a; McMahan et al., 2017; Mohri et al., 2019; Kairouz et al., 2019). A centralized model, here referred to as a server model, is then trained, without ever transmitting client data over the network, thereby providing some basic levels of data privacy and security.

Two important settings are distinguished in Federated learning (Kairouz et al., 2019, Table 1): the *cross-device* and the *cross-silo* settings. The cross-silo setting corresponds to a relatively small number of reliable clients, typically organizations, such as medical or financial institutions. In contrast, in the *cross-device* federated learning setting, the number of clients may be extremely large and include, for example, all 3.5 billion active android phones (Holst, 2019). Thus, in that setting, we may never make even a single pass over the entire clients' data during training. The cross-device setting is further characterized by resource-poor clients communicating over a highly unreliable network. Together, the essential features of this setting give rise to unique challenges not present in the cross-silo setting. Here, we are interested in the cross-device setting, for which we will formalize and study stochastic optimization algorithms.

The de facto standard algorithm for this setting is FEDAVG (McMahan et al., 2017), which performs multiple SGD updates on the available clients, before communicating to the server. While this approach can reduce the total amount of communication required, performing multiple steps on the same client can lead to 'over-fitting' to its atypical local data, a phenomenon known as *client drift* (Karimireddy et al., 2020). Furthermore, algorithmic innovations such as momentum (Sutskever et al., 2013; Cutkosky and Orabona, 2019), adaptivity (Kingma and Ba, 2014; Zaheer et al., 2018; Zhang et al., 2019), and clipping (You et al., 2017; 2019; Zhang et al., 2020) are critical to the success of deep learning applications and need to be incorporated into the client updates, replacing the SGD update of FEDAVG. Perhaps due to such deficiencies, there exists a large gap in performance between the centralized setting, where data is centrally collected on the server, and the federated setting (Zhao et al., 2018; Hsieh et al., 2019; Hsu et al., 2019; Karimireddy et al., 2020).

To overcome such deficiencies, we propose a new framework, MIME, that mitigates client drift and adapts arbitrary centralized optimization algorithms, e.g. SGD with momentum or Adam, to the federated setting. In each local client update, MIME uses global statistics, e.g. momentum, and an

SVRG-style correction to mimic the updates of the centralized algorithm run on i.i.d. data. These global statistics are computed only at the server level and kept fixed throughout the local steps, thereby avoiding a bias due to the atypical local data of any single client.

**Contributions.** We summarize our main results below.

- We formalize the cross-device federated learning problem, and propose a new framework MIME that can adapt arbitrary centralized algorithms to this setting.
- We prove that incorporating *server momentum* into each *local* client update reduces client drift and leads to optimal statistical rates.
- Further, we quantify the usefulness of performing multiple local updates on a single client by carefully tracking the bias (client-drift) introduced. This is the first analysis showing improved rates by taking additional multiple steps for general smooth functions.
- Finally, we also propose a simpler variant, MIMELITE, with an empirical performance similar to MIME. We report the results of thorough experimental analysis demonstrating that both MIME and MIMELITE are faster than FEDAVG.

**Related work.** *Analysis of FedAvg:* Much of the recent work in federated learning has focused on analyzing FEDAVG. For identical clients, FEDAVG coincides with parallel SGD, for which Zinkevich et al. (2010) derived an analysis with asymptotic convergence. Sharper and more refined analyses of the same method, sometimes called local SGD, were provided by Stich (2019), and more recently by Stich and Karimireddy (2019), Patel and Dieuleveut (2019), Khaled et al. (2020), and Woodworth et al. (2020b), for identical functions. Their analysis was extended to heterogeneous clients in (Wang et al., 2019; Yu et al., 2019b; Karimireddy et al., 2020; Khaled et al., 2020; Koloskova et al., 2020). Charles and Konečnỳ (2020) derived a tight characterization of FedAvg with quadratic functions and demonstrated the sensitivity of the algorithm to both client and server step sizes. Matching upper and lower bounds were recently given by Karimireddy et al. (2020) and Woodworth et al. (2020a) for general functions, proving that FEDAVG can be slower than even SGD for heterogeneous data, due to the *client-drift*.

*Comparison to SCAFFOLD:* For the cross-silo setting where the number of clients is relatively low, Karimireddy et al. (2020) proposed the SCAFFOLD algorithm, which uses control-variates (similar to SVRG) to correct for client drift. However, their algorithm crucially relies on *stateful clients* which repeatedly participate in the training process. In contrast, we focus on the cross-device setting where clients may be visited only once during training and where they are *stateless*. This is akin to the difference between the finite-sum and stochastic settings in traditional centralized optimization.

*Improvements to FedAvg:* Hsu et al. (2019) and Wang et al. (2020c) observed that using *server momentum* significantly improves over vanilla FEDAVG. This idea was generalized by Reddi et al. (2020), who replaced the server update with an arbitrary optimizer, e.g. Adam. However, these methods only modify the server update while using SGD for the client updates. MIME, on the other hand, ensures that every *local client update* resembles the optimizer e.g. MIME would apply momentum in every client update and not just at the server level. Beyond this, Li et al. (2018) proposed to add a regularizer to ensure client updates remain close. However, its usefulness is unclear (cf. Fig. 5, Karimireddy et al., 2020; Wang et al., 2020b). Other orthogonal directions which can be combined with MIME include tackling computation heterogeneity, where some clients perform many more updates than others (Wang et al., 2020b), improving fairness by modifying the objective (Mohri et al., 2019; Li et al., 2019), incorporating differential privacy (Geyer et al., 2017; Agarwal et al., 2018; Thakkar et al., 2020), Byzantine adversaries (Pillutla et al., 2019; Wang et al., 2020a; He et al., 2020a), secure aggregation (Bonawitz et al., 2017; He et al., 2020b), etc. We refer the reader to the extensive survey by Kairouz et al. (2019) for additional discussion.

## 2 PROBLEM SETUP

This section formalizes the problem of cross-device federated learning. We first examine some key challenges of this setting (cf. Kairouz et al., 2019) to ensure our formalism captures the difficulty:

1. Communication cost between the server and the clients is a major concern and the source of bottleneck in federated learning; thus, a key metric for optimization in this setting is the number of communication rounds.

2. Each client is likely to participate at most once, due to the extremely large number of clients; furthermore, each individual client may have very little data of its own.
3. There may be a wide heterogeneity or *non-i.i.d.-ness* due to the difference of data distributions for the clients.

Thus, our objective will be to minimize the following quantity within the fewest number of client-server communication rounds:

$$f(\boldsymbol{x}) = \mathbb{E}_{i \sim \mathcal{D}} \Big[ f_i(\boldsymbol{x}) := \frac{1}{n_i} \sum_{\nu=1}^{n_i} f_i(\boldsymbol{x}; \zeta_{i,\nu}) \Big]. \tag{1}$$

Here, $f_i$ denotes the loss function of client $i$ and $\{\zeta_{i,1}, \ldots, \zeta_{i,n_i}\}$ its local data. Since the number of clients is extremely large, while size of each local data is rather modest, we represent the former as an expectation and the latter as a finite sum. In each round, the algorithm samples a subset of clients (of size $S$) and performs some updates to the server model. There is some inherent tension between the second and the third challenge outlined above: if there exists a client with arbitrarily different data whom we may never encounter during training, then there is no hope to actually minimize $f$. Thus for (1) to be tractable, it is **necessary** to assume bounded dissimilarity between different $f_i$.

**(A1)** $G^2$-**BGD** or bounded gradient dissimilarity: there exists $G \geq 0$ such that

$$\mathbb{E}_{i \sim \mathcal{D}}[\|\nabla f_i(\boldsymbol{x}) - \nabla f(\boldsymbol{x})\|^2] \leq G^2, \ \forall \boldsymbol{x}.$$

Next, we also characterize the variance in the Hessians. Note that if $f_i(\cdot; \zeta)$ is $L$-smooth, (A2) is always satisfied with $\delta \leq 2L$ and hence is more of a *definition* rather than an assumption. Note that however, in realistic examples we expect the clients to be similar and hence that $\boldsymbol{\delta} \ll \boldsymbol{L}$.

**(A2)** $\delta$-**BHD** or bounded Hessian dissimilarity: Almost surely, $f$ is $\delta$-weakly convex i.e. $\nabla^2 f_i(\boldsymbol{x}) \succeq -\delta I$ and the loss function of any client $i$ satisfies

$$\|\nabla^2 f_i(\boldsymbol{x}; \zeta) - \nabla^2 f(\boldsymbol{x})\| \leq \delta, \ \forall \boldsymbol{x}.$$

In addition, we assume that $f(\boldsymbol{x})$ is bounded from below by $f^\star$ and is $L$-smooth, as is standard.

## 3 Using momentum to reduce client drift

In this section we examine the tension between reducing communication by running multiple client updates each round, and degradation in performance due to client drift (Karimireddy et al., 2020). To simplify the discussion, we assume a single client is sampled each round and that clients use full-batch gradients.

**Server-only approach.** A simple way to avoid the issue of client drift is to take no local steps. We sample a client $i \sim \mathcal{D}$ and run SGDm with momentum parameter $\beta$ and step size $\eta$:

$$\begin{aligned} \boldsymbol{x}_t &= \boldsymbol{x}_{t-1} - \eta \left( (1-\beta) \nabla f_i(\boldsymbol{x}_{t-1}) + \beta \boldsymbol{m}_{t-1} \right), \\ \boldsymbol{m}_t &= (1-\beta) \nabla f_i(\boldsymbol{x}_{t-1}) + \beta \boldsymbol{m}_{t-1}. \end{aligned} \tag{2}$$

Here, the gradient $\nabla f_i(\boldsymbol{x}_t)$ is *unbiased* i.e. $\mathbb{E}[\nabla f_i(\boldsymbol{x}_t)] = \nabla f(\boldsymbol{x}_t)$ and hence we are guaranteed convergence. However, this strategy can be communication-intensive and we are likely to spend all our time waiting for communication with very little time spent on computing the gradients.

**FedAvg approach.** To reduce the overall communication rounds required, we need to make more progress in each round of communication. Starting from $\boldsymbol{y}_0 = \boldsymbol{x}_{t-1}$, FEDAVG (McMahan et al., 2017) runs multiple SGD steps on the sampled client $i \sim \mathcal{D}$

$$\boldsymbol{y}_k = \boldsymbol{y}_{k-1} - \eta \nabla f_i(\boldsymbol{y}_{k-1}) \text{ for } k \in [K], \tag{3}$$

and then a pseudo-gradient $\tilde{\boldsymbol{g}}_t = -(\boldsymbol{y}_K - \boldsymbol{x}_t)$ replaces $\nabla f_i(\boldsymbol{x}_{t-1})$ in the SGDm algorithm (2). This is referred to as server-momentum since it is computed and applied only at the server level (Hsu et al., 2019). However, such updates give rise to *client-drift* resulting in performance worse than the naive server-only strategy (2). This is because by using multiple local updates, (3) starts over-fitting to the local client data, optimizing $f_i(\boldsymbol{x})$ instead of the actual global objective $f(\boldsymbol{x})$. The net

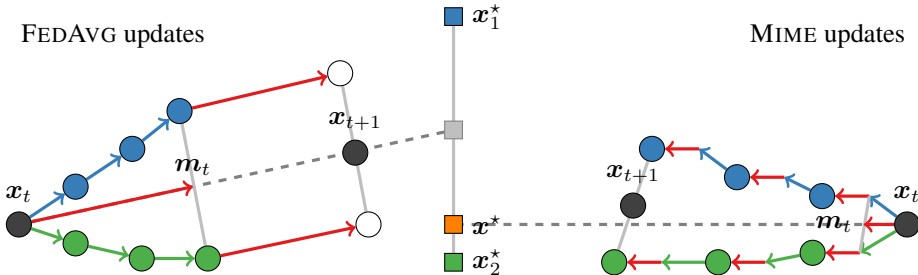

Figure 1: Client-drift in FEDAVG (left) and MIME (right) is illustrated for 2 clients with 3 local steps and momentum parameter $\beta = 0.5$. The local SGD updates of FEDAVG (shown using arrows for client 1 and client2) move towards the average of client optima $\frac{x_1^\star + x_2^\star}{2}$ which can be quite different from the true global optimum $x^\star$. Server momentum only speeds up the convergence to the wrong point in this case. In contrast, MIME uses unbiased momentum and applies it locally at every update. This keeps the updates of MIME closer to the true optimum $x^\star$.

effect is that FEDAVG moves towards an incorrect point (see Fig 1, left). If $K$ is sufficiently large, approximately

$$y_K \rightsquigarrow x_i^\star, \quad \text{where } x_i^\star := \arg\min_x f_i(x)$$

$$\Rightarrow \mathbb{E}_{i \sim \mathcal{D}}[\tilde{g}_t] \rightsquigarrow (x_t - \mathbb{E}_{i \sim \mathcal{D}}[x_i^\star]).$$

Further, the server momentum is based on $\tilde{g}_t$ and hence is also biased. Thus, it cannot correct for the client drift. We next see how a different way of using momentum can mitigate client drift.

**Mime approach.** FEDAVG experiences client drift because both the momentum and the client updates are biased. To fix the former, we compute momentum using only global statistics as in (2):

$$m_t = (1 - \beta)\nabla f_i(x_{t-1}) + \beta m_{t-1}. \tag{4}$$

To reduce the bias in the local updates, we will apply this unbiased momentum every step:

$$y_k = y_{k-1} - \eta((1 - \beta)\nabla f_i(y_{k-1}) + \beta m_{t-1}) \quad \text{for } k \in [K]. \tag{5}$$

Note that the momentum term is kept fixed during the local updates i.e. there is no local momentum used, only global momentum is applied locally. Since $m_{t-1}$ is a moving average of unbiased gradients computed over multiple clients, it intuitively is a good approximation of the general direction of the updates. By taking a convex combination of the local gradient with $m_{t-1}$, the update (5) is potentially also less biased. In this way MIME combines the communication benefits of taking multiple local steps and prevents client-drift (see Fig 1, right). Sec. C makes this intuition precise.

## 4 MIME FRAMEWORK

In this section we describe how to adapt arbitrary centralized algorithms (and not just SGDm) to the federated learning problem (1) while ensuring there is no client-drift. Algorithm 1 describes two variants MIME and MIMELITE, which consists of three components i) a base algorithm we are trying to mimic, ii) how we compute the global statistics, and iii) the local client updates.

**Base algorithm.** We assume the centralized base algorithm we are imitating can be decomposed into two steps: an *update step* $\mathcal{U}$ which updates the parameters $x$, and a *statistics step* $\mathcal{V}(\cdot)$ which keeps track of global statistics $s$. Each step of the base algorithm $\mathcal{B} = (\mathcal{U}, \mathcal{V})$ uses a gradient $g$ to

$$x \leftarrow x - \eta \mathcal{U}(g, s),$$
$$s \leftarrow \mathcal{V}(g, s). \tag{BASEALG}$$

$\mathcal{V}$ may track multiple statistics which we represent collectively as $s$. While SGDm (2) is clearly of this form, Appendix **??** shows this for other algorithms like Adam, etc.

---

**Algorithm 1** **Mime** and **MimeLite**

---

   **input:** initial $\boldsymbol{x}$ and $\boldsymbol{s}$, learning rate $\eta$ and base algorithm $\mathcal{B} = (\mathcal{U}, \mathcal{V})$
   **for** each round $t = 1, \cdots, T$ **do**
      sample subset $\mathcal{S}$ of clients
      **communicate** $(\boldsymbol{x}, \boldsymbol{s})$ to all clients $i \in \mathcal{S}$
      **communicate** $\boldsymbol{c} \leftarrow \frac{1}{|\mathcal{S}|} \sum_{j \in \mathcal{S}} \nabla f_j(\boldsymbol{x})$ (only for Mime)
      **on client** $i \in \mathcal{S}$ **in parallel do**
         initialize local model $\boldsymbol{y}_i \leftarrow \boldsymbol{x}$
         **for** $k = 1, \cdots, K$ **do**
            sample mini-batch $\zeta$ from local data
            $\boldsymbol{y}_i \leftarrow \boldsymbol{y}_i - \eta \mathcal{U}(\nabla f_i(\boldsymbol{y}_i; \zeta) - \nabla f_i(\boldsymbol{x}; \zeta) + \boldsymbol{c}, \boldsymbol{s})$ **(Mime)**
            $\boldsymbol{y}_i \leftarrow \boldsymbol{y}_i - \eta \mathcal{U}(\nabla f_i(\boldsymbol{y}_i; \zeta), \boldsymbol{s})$ **(MimeLite)**
         **end for**
         compute full local-batch gradient $\nabla f_i(\boldsymbol{x})$
         **communicate** $(\boldsymbol{y}_i, \nabla f_i(\boldsymbol{x}))$
      **end on client**
      $\boldsymbol{s} \leftarrow \mathcal{V}\left(\frac{1}{|\mathcal{S}|} \sum_{i \in \mathcal{S}} \nabla f_i(\boldsymbol{x}), \ \boldsymbol{s}\right)$ (update optimization statistics)
      $\boldsymbol{x} \leftarrow \frac{1}{|\mathcal{S}|} \sum_{i \in \mathcal{S}} \boldsymbol{y}_i$ (update server parameters)
   **end for**

---

**Compute statistics globally, apply locally.** When updating the statistics of the base algorithm, we use only the gradient computed at the server parameters. Further, they remain fixed throughout the local updates of the clients. This ensures that these statistics remain unbiased and representative of the global function $f(\cdot)$. At the end of the round, the server performs

$$\boldsymbol{s} \leftarrow \mathcal{V}\left(\frac{1}{|\mathcal{S}|} \sum_{i \in \mathcal{S}} \nabla f_i(\boldsymbol{x}), \ \boldsymbol{s}\right), \quad \text{where } \nabla f_i(\boldsymbol{x}) = \frac{1}{n_i} \sum_{\nu=1}^{n_i} \nabla f_i(\boldsymbol{x}; \zeta_{i,\nu}). \tag{STATS}$$

Note that we use full-batch gradients computed at the server parameters $\boldsymbol{x}$, not client parameters $\boldsymbol{y}_i$.

**Local client updates.** Each client $i \in \mathcal{S}$ performs $K$ updates using $\mathcal{U}$ of the base algorithm and a minibatch gradient. There are two variants possible corresponding to MIME and MIMELITE differentiated using colored boxes. Starting from $\boldsymbol{y}_i \leftarrow \boldsymbol{x}$, repeat the following $K$ times

$$\boldsymbol{y}_i \leftarrow \boldsymbol{y}_i - \eta \mathcal{U}(\boldsymbol{g}, \boldsymbol{s}), \text{ where}$$
$$\boldsymbol{g} \leftarrow \boxed{\nabla f_i(\boldsymbol{y}_i; \zeta) - \nabla f_i(\boldsymbol{x}; \zeta) + \frac{1}{|\mathcal{S}|} \sum_{j \in \mathcal{S}} \nabla f_j(\boldsymbol{x})} \text{ or } \boxed{\nabla f_i(\boldsymbol{y}_i; \zeta)}. \tag{CLTSTEP}$$

MIMELITE simply uses the local minibatch gradient whereas MIME uses an SVRG style correction (Johnson and Zhang, 2013). This is done to reduce the noise from sampling a local mini-batch. While this correction yields faster rates in theory (and in practice for convex problems), in deep learning applications we found that MIMELITE closely matches the performance of MIME.

Finally, there are two modifications made in practical FL: we weigh all averages across the clients by the number of datapoints $n_i$, and we perform $K$ epochs instead of $K$ steps (McMahan et al., 2017). The former modifies the objective (1) with $f_i$ being weighted by $n_i$, and the latter has been empirically observed to perform better, but lacks strong justification (Wang et al., 2020b).

## 5 THEORETICAL ANALYSIS OF MIME

Table 1 summarizes the rates of server-only methods, FEDAVG, SCAFFOLD and MIME (new results highlighted in blue). Our theory focuses on MIME with base algorithm of SGD with momentum based variance reduction[1] (MimeMVR) since it obtains optimal rates.

---

[1]The momentum based variance reduction (MVR), introduced by Cutkosky and Orabona (2019), is a modification of the standard SGDm algorithm to make it amenable to analysis. All our theory uses MVR, while our experiments use SGDm.

Table 1: Number of communication rounds required to reach $\|\nabla f(\boldsymbol{x})\|^2 \leq \epsilon$ for $L$-smooth functions (log factors are ignored) with $S$ clients sampled each round. $G^2$ bounds the gradient dissimilarity (A1), and $\delta$ bounds the Hessian dissimilarity (A2). FEDAVG is slower than the server-only methods due to additional drift terms. Convergence of SCAFFOLD depends on the total number of clients $N$ which is potentially infinite. MIME matches the optimal statistical rates (first term in the rates) of the server-only methods while improving the optimization (second) term (typically $\delta \ll L$).

| Algorithm | Non-convex | $\mu$-Strongly convex |
|---|---|---|
| SERVER-ONLY | | |
|     SGD (Ghadimi and Lan, 2013) | $\frac{G^2}{S\epsilon^2} + \frac{L}{\epsilon}$ | $\frac{G^2}{\mu S\epsilon} + \frac{L}{\mu}$ |
|     MVR (Cutkosky and Orabona, 2019) | $\left(\frac{G}{\sqrt{S}\epsilon}\right)^{\frac{3}{2}} + \frac{L}{\epsilon}$ | – |
| FEDAVG [1] | | |
|     FedSGD (Karimireddy et al., 2020) | $\frac{G^2}{S\epsilon^2} + \frac{G}{\epsilon^{3/2}} + \frac{L}{\epsilon}$ | $\frac{G^2}{\mu S\epsilon} + \frac{G}{\mu\sqrt{\epsilon}} + \frac{L}{\mu}$ |
| SCAFFOLD [2] (Karimireddy et al., 2020) | $\left(\frac{N}{S}\right)^{\frac{2}{3}}\frac{L}{\epsilon}$ | $\frac{N}{S} + \frac{L}{\mu}$ |
| MIME [3] | | |
|     MimeSGD | $\frac{G^2}{S\epsilon^2} + \frac{\delta}{\epsilon}$ | $\frac{G^2}{\mu S\epsilon} + \frac{\delta}{\mu}$ |
|     MimeMVR | $\left(\frac{G}{\sqrt{S}\epsilon}\right)^{\frac{3}{2}} + \frac{\delta}{\epsilon}$ | – |
| Lower bound (Arjevani et al., 2019) | $\Omega\left(\frac{G}{\sqrt{S}\epsilon}\right)^{\frac{3}{2}}$ | $\Omega\left(\frac{G^2}{S\epsilon}\right)$ |

[1] Requires $K \geq \frac{\sigma^2}{G^2}$ number of local updates with within-client variance of $\sigma^2$.
[2] In cross-device FL, the total number of clients ($N$) can be of the same order as number of rounds (since we only make few passes over all clients), or even $\infty$, making the bounds vacuous.
[3] Requires $K \geq L/\delta$ number of local updates.

**Theorem I.** *For $L$-smooth $f$ with $G^2$ gradient dissimilarity (A1), $\delta$ Hessian dissimilarity (A2) and $F := (f(\boldsymbol{x}^0) - f^\star)$, let us run MimeMVR for $T$ rounds and generate an output $\boldsymbol{x}^{out}$. This output satisfies $\mathbb{E}\|\nabla f(\boldsymbol{x}^{out})\|^2 \leq \epsilon$ under the following conditions*

- **PL-Strongly convex without momentum:** *for $\eta = \tilde{\mathcal{O}}\left(\min\left(\frac{1}{\delta K + \mu K + L}, \frac{1}{\mu T}\right)\right)$, $\beta = 0$, and*
$$T = \tilde{\mathcal{O}}\left(\frac{LG^2}{\mu S\epsilon} + \frac{L + \delta K}{\mu K}F\log\left(\frac{1}{\epsilon}\right)\right),$$

- **Non-convex without momentum:** *for $\eta = \mathcal{O}\left(\min\left(\frac{1}{\delta K + L}, \left(\frac{SF}{G^2 TK^2}\right)^{1/2}\right)\right)$, $\beta = 0$, and*
$$T = \mathcal{O}\left(\frac{LG^2 F}{S\epsilon^2} + \frac{(L + \delta K)F}{\epsilon K}\right),$$

- **Non-convex with momentum:** *for $\eta = \mathcal{O}\left(\min\left(\frac{1}{\delta K + L}, \left(\frac{SF}{G^2 TK^3}\right)^{1/3}\right)\right)$, $\beta = 1 - \mathcal{O}\left(\frac{\delta^2}{(TG^2)^{2/3}}\right)$,*
$$T = \mathcal{O}\left(\left(\frac{(1+\delta)G^2 F}{S\epsilon^2}\right)^{3/4} + \frac{(L + \delta K)F}{\epsilon K}\right).$$

The expectation in $\mathbb{E}\|\nabla f(\boldsymbol{x}^{\text{out}})\|^2 \leq \epsilon$ is taken both over the sampling of the clients during the running of the algorithm, the sampling of the mini-batches in local updates, and the choice of $\boldsymbol{x}^{\text{out}}$ (which is chosen randomly from the client iterates $\boldsymbol{y}_i$ as described in the Appendix).

Table 1 shows that the rate (ignoring constants) of FEDAVG on non-convex functions is $\left(\frac{G^2}{S\epsilon^2} + \frac{G}{\epsilon^{3/2}} + \frac{L}{\epsilon}\right)$. This is *slower* than simply running SGD which obtains a rate of $\left(\frac{G^2}{S\epsilon^2} + \frac{L}{\epsilon}\right)$. In contrast, MimeSGD obtains a rate of $\left(\frac{G^2}{S\epsilon^2} + \frac{\delta}{\epsilon}\right)$ where $\delta \ll L$ thus improving upon both SGD and FedAvg. While asymptotically these three rates may seem equivalent, in machine learning we care about low accuracy settings where $\epsilon$ is not too small (Bottou, 2010) and so the lower order terms matter, as

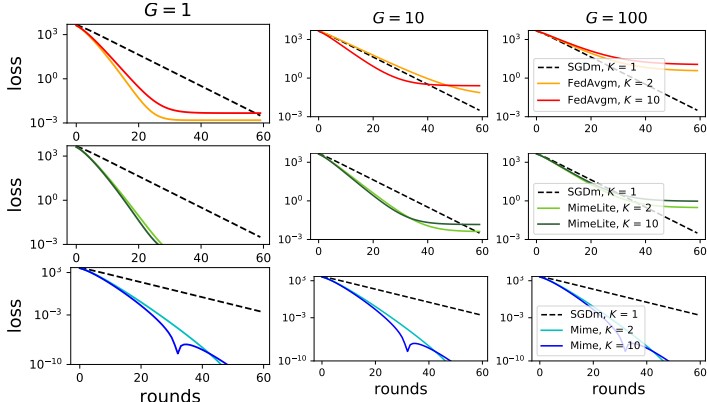

Figure 2: **SGDm** (dashed black), **FedSGDm** (top), **MimeLiteSGDm** (middle), and **MimeSGDm** (bottom) on simulated data, all with momentum ($\beta = 0.5$). FedAvg gets slower as the gradient-dissimilarity ($G$) increases (to the right). MimeLite shows a similar pattern, but is consistently better than FedAvg. Mime is significantly faster than both and is unaffected by heterogeneity ($G$).

also additionally evidenced by our experiments (Sec. 6). We can also compare with SCAFFOLD (Karimireddy et al., 2020) which obtains a rate of $\left(\frac{N}{S}\right)^{\frac{2}{3}} \frac{L}{\epsilon}$ where $N$ is the total number of clients. While asymptotically this is a faster rate, $N$ in the cross-device setting is potentially infinite or at least comparable to the total number of training rounds, making these bounds vacuous. This too is reflected in our experiments (Fig. 5).

Finally, by incorporating momentum based variance reduction, MimeMVR matches the lower bound of $\Omega\left(G/\sqrt{S}\epsilon\right)^{\frac{3}{2}}$ by Arjevani et al. (2019). The momentum $\beta$ used in this case is of the order of $(1 - \mathcal{O}(TG^2)^{-2/3})$ i.e. as $T$ increases, our momentum parameter asymptotically approaches 1. In contrast, previous analyses of distributed momentum (e.g. Yu et al. (2019a)) prove rates of the form $\frac{G^2}{S(1-\beta)\epsilon^2}$, which are worse than that of standard SGD by a factor of $\frac{1}{1-\beta}$. Thus, ours is the first result which theoretically showcases the usefulness of using large momentum in distributed and federated learning. Our theory suggests that the momentum parameter should be increased if $G$ increases i.e. as the clients become more heterogeneous, there is stronger client-drift and hence we need more momentum to compensate.

Our analysis is is highly non-trivial and involves three crucial ingredients: i) computing the momentum at the server level to ensure that it remains unbiased and then applying it locally during every client update to reduce variance, ii) carefully keeping track of the bias introduced via additional local steps, and iii) an SVRG correction to allow using mini-batches. Our experiments (Sec. 6) verify that the first two theoretical insights are indeed applicable in deep learning settings as well, whereas the latter seems to matter more in convex settings. See App. C where we make this discussion more concrete and Appendices F–G for detailed proofs and theorem statements.

## 6 EXPERIMENTAL ANALYSIS

We run experiments on simulated and real datasets to confirm our theory. Our main findings are i) MIME outperforms FEDAVG across all settings, ii) its SVRG correction is useful for convex problems, and iii) momentum significantly improves performance for non-convex problems.

We consider four algorithms: SERVER-ONLY, FEDAVG, MIME, and MIMELITE. Each of these adapt base optimizers SGD, SGDm, and Adam. The SERVER-ONLY method computes a full batch gradient on each of the sampled clients and uses their aggregate directly in the base optimizer (akin to (2)). For FEDAVG, we follow Reddi et al. (2020) who run multiple epochs of SGD on each client sampled, and then aggregate the net client updates. This aggregated update is used as a pseudo-gradient in the base optimizer (called server optimizer). The learning rate for the server optimizer is fixed to 1 as in (Wang et al., 2020c). This is done to ensure all algorithms have the same number of hyper-parameters. Finally, MIME and MIMELITE follow Algorithm 1 and also run a fixed number of epochs on the client. Aggregation is weighted by the number of samples on the clients.

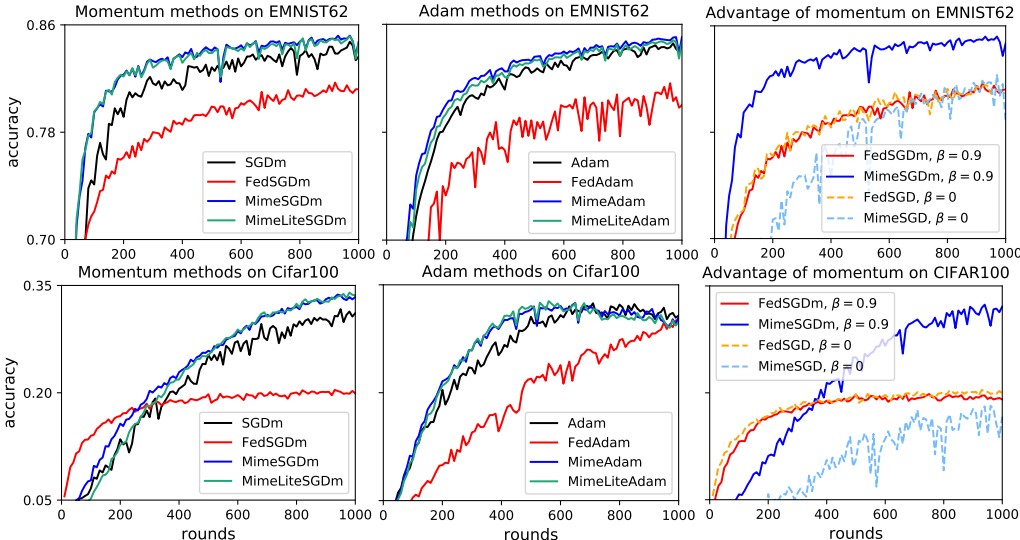

Figure 3: **Server-only**, **FedAvg**, **Mime**, and **MimeLite** with SGDm (left) and Adam (middle) run on (top) EMNIST62 and a 2 hidden layer (300u-100) MLP and (bottom) Resnet20 run on Cifar100. Mime and MimeLite have very similar performance and are consistently the best. FedAvg is even worse than the server-only baselines. Also, Mime makes better use of momentum than FedAvg, with a large increase in performance (right).

### 6.1 SIMULATED CONVEX EXPERIMENTS

Our simulated experiments use two clients each with a simple scalar quadratic loss, as in (Karim-ireddy et al., 2020). We use full-batch gradients with both clients participating every round. The simulated data has Hessian dissimilarity $\delta = 1$ (A2) and smoothness $L = 2$. We vary the gradient dissimilarity (A1) as $G \in [1, 10, 100]$. All the algorithms use momentum with $\beta = 0.5$ and their learning rates were tuned up to a tolerance of 5E-3 to ensure lowest loss after 60 rounds. The results are collected in Fig. 2. When $G$ is small, we see that FEDAVG can outperform the SERVER-ONLY (SGDm) baseline, though its loss quickly plateaus. On increasing $G$, FEDAVG becomes even slower. MIMELITE differs from FEDAVG only in how the momentum is used. In all settings, it slightly out-performs FEDAVG though even it sees a substantial slow down as we increase $G$. This reflects our theory which predicts that for convex cases, momentum does not give significant gains. MIME, on the other hand, is substantially faster than all other methods and is even unaffected by changing $G$. Thus, in this simple convex setting, the SVRG correction completely eliminates client drift.

### 6.2 REAL WORLD EXPERIMENTS

We run real world deep learning experiments on EMNIST62 with a 2 layer MLP model and on Cifar100 with ResNet20, both accessed through Tensorflow Federated (TFF, 2020a). All methods run 10 local epochs, batch size 20, and the learning rates for all methods were individually tuned. We refer to Appendix A for additional details and results. Fig 3 shows the results.

**Mime > MimeLite > Server-only > FedAvg.** Mime and MimeLite have the best performance. FedAvg is slower than even the naive server-only methods which make no local updates. This perfectly mirrors our theory that Mime > server-only > FedAvg. The performance of MimeLite is because SVRG correction may not be necessary in deep learning (Defazio and Bottou, 2019).

**With momentum > without momentum.** Fig. 3 (right) examines the impact of momentum on FedAvg and Mime. Momentum slightly improves the performance of FedAvg, whereas it has a significant impact on the performance of Mime. This is also in line with our theory and confirms that Mime's strategy of applying it locally at every client update makes better use of momentum.

**Fixed statistics > updated statistics.** Finally, we check how the performance of Mime changes if instead of keeping the momentum fixed throughout a round, we let it change. The momentum is reset at the end of the round ignoring the changes the clients make to it. Appendix B.1 shows that this consistently *worsens* the performance, confirming that it is better to keep the statistics fixed.

Together, the above observations validate all aspects of Mime (and MimeLite) design: compute statistics at the server level, and apply them unchanged at every client update.

## 7 CONCLUSION

Our work initiated a formal study of the cross-device federated learning problem. We argued that the natural heterogeneity among the clients gives rise to client drift and significantly hampers the performance of approaches such as FEDAVG. We then showed how momentum can be an excellent tool to overcome this client drift if used correctly. Based on this observation, we introduced a new framework MIME which not only overcomes client drift, but also adapts arbitrary centralized algorithms such as Adam to the federated setting without any additional hyper-parameters. We demonstrated the superiority of MIME via strong convergence guarantees and empirical evaluations.

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

# Appendix

## A  EXPERIMENTAL SETUP

### A.1  DESCRIPTION OF DATASETS AND TASKS

We perform 4 tasks over 3 datasets: i) On the EMNIST62 (extended MNIST) dataset (Caldas et al., 2018b) we run a convex logistic regression model, a fully connected MLP with 2 hidden layers (300u-100), and convolution model with two CNN layers and two dense layers with dropout. ii) we also run a ResNet20 on CIFAR100.

In all cases we report the top-1 test accuracy in our experiments. EMNIST uses the metadata indicating the original author of the characters to separate them into multiple clients yielding naturally partitioned dataset. Table 2 summarizes the statistics about the different datasets. Note that the average number of rounds a client participates in (computed as sampled clients×number of rounds/number of clients) provides an indication of how much of the training data is seen with SHAKESPEARE being closest to the cross-silo setting and STACKOVERFLOW representing the most cross-device in nature.

Table 2: Details about the datasets used and experiment setting.

|                                        | EMNIST62 | CIFAR100 |
| -------------------------------------- | -------- | -------- |
| Clients                                | 3,400    | 500      |
| Examples                               | 671,585  | 50,000   |
| Sampled clients                        | 20       | 20       |
| Batch size                             | 20       | 20       |
| Number of epochs                       | 10       | 10       |
| Avg. rounds each client participates   | 5.9      | 40       |

We use Tensorflow federated datasets (TFF, 2020a) to generate the datasets. Our federated learning simulation code is written in Jax (Frostig et al., 2018). Our Resnet18 model is based off of (Haiku, 2020) (Resnet v2), and following (Hsieh et al., 2019; Reddi et al., 2020) we replace batch norm with group norm with 2 groups. Black and white was reversed in EMNIST62 (i.e. subtracted from 1) to make them similar to MNIST. CIFAR100 used the usual pre-processing (normalization and centering), and data augmentation (random crop and horizontal flipping) following (kuangliu, 2020 (accessed June 4, 2020)).

### A.2  PRACTICALITY OF EXPERIMENTS

In the experiments we only cared about the number of communication rounds, ignoring that MIME actually needs twice the number of bits per round and that the SERVER-ONLY methods have a much smaller computational requirement. This is standard in the federated learning setting as introduced by McMahan et al. (2017) and is justified because most of the time in cross-device FL is spent in establishing connections with devices rather than performing useful work such as communication or computation. In other words, *latency* and not bandwidth or computation are critical in cross device FL. However, one can certainly envision cases where this is not true. Incorporating communication compression strategies (Suresh et al., 2017; Alistarh et al., 2017; Karimireddy et al., 2019) or client-model compression strategies (Caldas et al., 2018a; Frankle and Carbin, 2019; Hamer et al., 2020) into our MIME framework can potentially address such issues and are important future research directions.

As we already discussed previously, we believe both the datasets and the tasks being studied here are quite realistic in nature. We now discuss our choice of other parameters in the experiment setup (number of training rounds, sampled clients, batch-size, etc.) Each round of federated learning takes 2–3 mins in the real world and is relatively independent of the size of communication (Bonawitz et al., 2019) implying that training 1000 rounds takes **1.4–2 days**. This underscores the importance of ensuring that the algorithms for federated learning converge in as few rounds as possible, as well as have very easy to set default hyper-parameters. Thus in our experimental setup we keep

all parameters other than the learning rate to their default values. In practice, this learning rate can be set by set using a small dataset on the server (as in (Hard et al., 2020)). The choice of batch size being 10 was made both keeping in mind the limited memory available to each client as well as to match prior work. Finally, while we limit ourselves to sampling 20 workers per round due to computational constraints, in real world FL thousands of devices are often available for training simultaneously each round (Bonawitz et al., 2019). They also note that the probability of each of these devices being available has clear patterns and is far from uniform sampling. Conducting a large scale experimental study which mimics these alternate forms of heterogeneity is an important direction for future work.

### A.3    Hyperparameter search

We use the EMNIST62 with MLP model as a 'test-bed' for exploring different algorithms given it being both a representative task of cross-device FL as well as being computationally efficient. All plots reported are for this setting. A more fine-grained search over hyperparameters to report the final test accuracies is made over the rest of the tasks/datasets.

For all SGDm methods, we pick momentum $\beta = 0.9$. For Adam methods, we fix $\beta_1 = 0.9$, $\beta_2 = 0.99$, and $\varepsilon = 1 \times 10^{-3}$ similar to (Reddi et al., 2020). None of the algorithms use weight decay, clipping etc. The learning rate is then tuned to obtain the best test accuracy.

For all experiments, unless explicitly mentioned otherwise, the learning rate is searched over a grid of

$$\eta \in [1 \times 10^1, 1, 1 \times 10^{-1}, 1 \times 10^{-2}, 1 \times 10^{-3}, 1 \times 10^{-4}, 1 \times 10^{-5}].$$

### A.4    Comparison with previous results

As far as we are aware, (Reddi et al., 2020) is the only prior work which conducts a systematic experimental study of federated learning algorithms over multiple realistic datasets. The algorithms comparable across the two works (e.g. FedSGD, FedSGDm, and FedAdam) have qualitatively similar performance except with one exception: FedAdam consistently underperforms FedSGDm. We believe this difference is because (Reddi et al., 2020) additionally tune the server learning rate and the $\epsilon$ parameter for Adam. As we explain in Section A.2, we chose to keep these parameters to some default values to compare methods in the 'low-tuning' setting. We also point that while FedAdam struggles to perform in this setup, MimeAdam and MimeLiteAdam are very stable and even often outperform their SGDm counterparts. This is also the default/recommended behavior in TensorFlow Federated (TFF, 2020b) and (Wang et al., 2020c).

## B    Additional experiments

### B.1    Effect of changing statistics

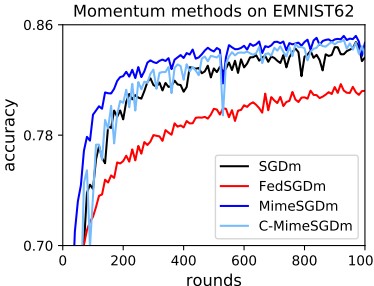 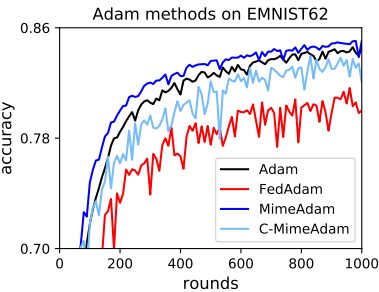

Figure 4: **Server-only**, **FedAvg**, **Mime**, and **C-Mime** with SGDm (left) and Adam (right) run on EMNIST62 with a 2 hidden layer (300u-100) MLP. C-Mime changes the statistics (momentum for SGDm, and first two moments for Adam) using the local client updates. These changes are discarded at the end of the round and the statistics are reset using only the server level gradients as in Mime. Clearly, C-Mime is always worse than Mime. This shows that adapting statistics during local client updates makes them too biased, and it best to keep them fixed during each round like Mime does.

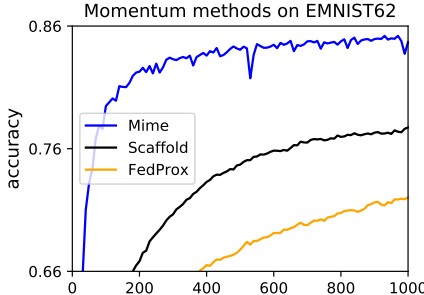

Figure 5: Comparison with Scaffold and FedProx for cross-device FL: **Mime**, **SCAFFOLD** and **FedProx** with SGDm run on EMNIST62 with a 2 hidden layer (300u-100) MLP. For FedProx and SCAFFOLD, in addition to tuning the learning rate, we search for the best server momentum $\beta \in [0, 0.9, 0.99]$. FedProx uses an additional regularizer $\mu$ which we search over $[0.1, 0.5, 1]$ (note that FedProx with $\mu = 0$ is the same as FedAvg). The best test accuracy (which are plotted here) was by $\beta = 0$ for both and $\mu = 0.1$ for FedProx. Note that FedProx is the slowest method here (in fact it is even slower than FedAvg). The additional regularizer does not seem to reduce client drift while still slowing down convergence (Karimireddy et al., 2020; Wang et al., 2020b). SCAFFOLD is also slower than Mime in this setup. This is because SCAFFOLD was designed for the cross-silo setting and not corss-device setting. The large number of clients ($N = 3.4k$) means that each client is on averaged visited less than 6 times during the entire training (20 clients per round for 1k rounds). Hence, the client control variate stored is quite stale (from about 200 rounds ago) which slows down the convergence. This perfectly reflects our theoretical understanding that when the number of clients $N$ is large relative to training rounds (which is true in the cross-device setting) SCAFFOLD is outperformed by MIME.

| | FEDAVG | | | MIME | | | MIMELITE | | |
|---|---|---|---|---|---|---|---|---|---|
| | SGD | SGDm | Adam | SGD | SGDm | Adam | SGD | SGDm | Adam |
| EMNIST62 logistic | 66.0 | 66.8 | 67.5 | 67.2 | 67.3 | **68.0** | 66.1 | 67.3 | **68.0** |
| EMNIST62 CNN | 84.9 | 85.4 | 85.7 | **86.0** | 85.6 | **86.0** | 85.0 | 85.4 | 85.7 |

Table 3: High tuning setting: final test accuracy (larger is better) with fully tuned hyper-parameters. For FEDAVG we searched over both the client and server learning rates, whereas for MIME and MIMELITE, we search only over client (base) learning rate. This search is performed over a grid (0.002, 0.005, 0.01, 0.02, 0.05, 0.1, 0.2, 0.4, 0.8). For momentum, we chose best of (0.9, 0.99) and for Adam, we varied $\beta_1$ in (0.9, 0.99), $\beta_2$ in (0.99, 0.999) and $\epsilon$ in (0.01, 0.001, 0.0001). In this setting as well MIME and MIMELITE outperform FEDAVG but with a smaller margin.

Table 4: Additional algorithmic details: Decomposing base algorithms into a parameter update ($\mathcal{U}$) and statistics tracking ($\mathcal{V}$).

| Algorithm | Tracked statistics $\boldsymbol{s}$ | Update step $\mathcal{U}$ | Tracking step $\mathcal{V}$ |
|---|---|---|---|
| SGD | – | $\boldsymbol{x} - \eta \boldsymbol{g}$ | – |
| SGDm | $\boldsymbol{m}$ | $\boldsymbol{x} - \eta((1-\beta)\boldsymbol{g} + \beta\boldsymbol{m})$ | $\boldsymbol{m} = (1-\beta)\boldsymbol{g} + \beta\boldsymbol{m}$ |
| RMSProp | $\boldsymbol{v}$ | $\boldsymbol{x} - \frac{\eta}{\epsilon + \sqrt{\boldsymbol{v}}}\boldsymbol{g}$ | $\boldsymbol{v} = (1-\beta)\boldsymbol{g}^2 + \beta\boldsymbol{v}$ |
| Adam | $\boldsymbol{m}, \boldsymbol{v}$ | $\boldsymbol{x} - \frac{\eta}{\epsilon + \sqrt{\boldsymbol{v}}}((1-\beta_1)\boldsymbol{g} + \beta_1\boldsymbol{m})$ | $\boldsymbol{m} = (1-\beta_1)\boldsymbol{g} + \beta_1\boldsymbol{m}$ $\boldsymbol{v} = (1-\beta_2)\boldsymbol{g}^2 + \beta_2\boldsymbol{v}$ |

## C  PROOF OVERVIEW

In this section, we give proof sketches of the main components of Theorem I: i) how momentum reduces the effect of client drift, ii) how local steps can take advantage of Hessian similarity, and iii) why the SVRG correction improves constants.

**Improving the statistical term via momentum.** Note that the statistical (first) term in Theorem I without momentum ($\beta = 0$) for the convex case is $\frac{LG^2}{\mu S \epsilon}$. This is (up to constants) optimal and cannot be improved. For the non-convex case however using $\beta = 0$ gives the usual rate of $\frac{LG^2}{S\epsilon^2}$. However, this can be improved to $\left(\frac{(1+\delta)G^2 F}{S\epsilon^2}\right)^{3/4}$ using momentum. This matches a similar improvement in the centralized setting (Cutkosky and Orabona, 2019; Tran-Dinh et al., 2019) and is in fact optimal (Arjevani et al., 2019). Let us examine why momentum improves the statistical term. Assume that we sample a single client $i_t$ in round $t$ and that we use full-batch gradients. Also let the local client update at step $k$ round $t$ be of the form

$$\boldsymbol{y} \leftarrow \boldsymbol{y} - \eta \boldsymbol{d}_k \,. \tag{6}$$

The ideal choice of update is of course $\boldsymbol{d}_k^\star = \nabla f(\boldsymbol{y})$ but however this is unattainable. Instead, MIME with momentum $\beta = 1 - a$ uses $\boldsymbol{d}_k^{\text{SGDm}} = \tilde{\boldsymbol{m}}_k \leftarrow a \nabla f_i(\boldsymbol{y}) + (1-a)\boldsymbol{m}_{t-1}$ where $\boldsymbol{m}_{t-1}$ is the momentum computed at the server. The variance of this update can then be bounded as

$$\mathbb{E}\|\tilde{\boldsymbol{m}}_k - \nabla f(\boldsymbol{y})\|^2 \lesssim a^2 \, \mathbb{E}\|\nabla f_{i_t}(\boldsymbol{y}) - \nabla f(\boldsymbol{y})\|^2 + (1-a)\,\mathbb{E}\|\boldsymbol{m}_{t-1} - \nabla f(\boldsymbol{y})\|^2$$
$$\approx a^2 G^2 + (1-a)\,\mathbb{E}\|\boldsymbol{m}_{t-1} - \nabla f(\boldsymbol{x}_{t-2})\|^2 \approx a G^2 \,.$$

The last step follows by unrolling the recursion on the variance of $\boldsymbol{m}$. We also assumed that $\eta$ is small enough that $\boldsymbol{y} \approx \boldsymbol{x}_{t-2}$. This way, momentum can reduce the variance of the update from $G^2$ to $(a G^2)$ by using past gradients computed on different clients. To formalize the above sketch requires slightly modifying the momentum algorithm similar to (Cutkosky and Orabona, 2019), and is carried out in Appendix G.

**Improving the optimization term via local steps.** The optimization (second) term in Theorem I for the convex case is $\frac{\delta K + L}{\mu K}$ and for the non-convex case (with or without momentum) is $\frac{\delta K + L}{\epsilon K}$. In contrast, the optimization term of the server-only methods is $L/\mu$ and $L/\epsilon$ respectively. Since in most cases $\delta \ll L$, the former can be significantly smaller than the latter. This rate also suggests that the best choice of number of local updates is $L/\delta$ i.e. we should perform more client updates when they have more similar Hessians. This generalizes results of (Karimireddy et al., 2020) from quadratics to all functions.

This improvement is due to a careful analysis of the *bias* in the gradients computed during the local update steps. Note that for client parameters $\boldsymbol{y}_{k-1}$, the gradient $\mathbb{E}[\nabla f_{i_t}(\boldsymbol{y}_{k-1})] \neq \mathbb{E}[\nabla f(\boldsymbol{y}_{k-1})]$ since $\boldsymbol{y}_{k-1}$ was also computed using the same loss function $f_{i_t}$. In fact, only the first gradient computed at $\boldsymbol{x}_{t-1}$ is unbiased. Dropping the subscripts $k$ and $t$, we can bound this bias as:

$$\mathbb{E}[\nabla f_i(\boldsymbol{y}) - \nabla f(\boldsymbol{y})] = \mathbb{E}[\underbrace{\nabla f_i(\boldsymbol{y}) - \nabla f_i(\boldsymbol{x})}_{\approx \nabla^2 f_i(\boldsymbol{x})(\boldsymbol{y}-\boldsymbol{x})} + \underbrace{\nabla f(\boldsymbol{x}) - \nabla f(\boldsymbol{y}_i)}_{\approx \nabla^2 f(\boldsymbol{x})(\boldsymbol{x}-\boldsymbol{y}_i)}] + \underbrace{\mathbb{E}_i[\nabla f_i(\boldsymbol{x})] - \nabla f(\boldsymbol{x})}_{=0 \text{ since unbiased}}$$
$$\approx \mathbb{E}[(\nabla^2 f_i(\boldsymbol{x}) - \nabla^2 f(\boldsymbol{x}))(\boldsymbol{y}_i - \boldsymbol{x})] \approx \delta \, \mathbb{E}[(\boldsymbol{y}_i - \boldsymbol{x})] \,.$$

Thus, the Hessian dissimilarity (A2) control the bias, and hence the usefulness of local updates. This intuition can be made formal using Lemma 3.

**Mini-batches via SVRG correction.** In our previous discussion about momentum and local steps, we assumed that the clients compute full batch gradients and that only one client is sampled per round. However, in practice a large number ($S$) of clients are sampled and further the clients use mini-batch gradients. The SVRG correction reduces this within-client variance since

$$\text{Var}\left(\nabla f_i(\boldsymbol{y}_i; \zeta) - \nabla f_i(\boldsymbol{x}; \zeta) + \tfrac{1}{|\mathcal{S}|} \sum_{i \in \mathcal{S}} \nabla f_i(\boldsymbol{x})\right) \lesssim L^2 \|\boldsymbol{y}_i - \boldsymbol{x}\|^2 + \frac{G^2}{S} \approx \frac{G^2}{S} \,.$$

Here, we used the smoothness of $f_i(\cdot; \zeta)$ and assumed that $\boldsymbol{y}_i \approx \boldsymbol{x}$ since we don't move too far within a single round. Thus, the SVRG correction allows us to use minibatch gradients in the local updates while still ensuring that the variance is of the order $G^2/S$.

## D  TECHNICALITIES

We examine some additional definitions and introduce some technical lemmas.

### D.1  ASSUMPTIONS AND DEFINITIONS

We make precise a few definitions and explain some of their implications. We first discuss the two assumptions on the dissimilarity between the gradients (A1) and the Hessians (A2). Loosely, these two quantities are an extension of the concepts of **variance** and **smoothness** which occur in centralized SGD analysis to the federated learning setting. Just as the variance and smoothness are completely orthogonal concepts, we can have settings where $G^2$ (gradient dissimilarity) is large while $\delta$ (Hessian dissimilarity) is small, or vice-versa.

Our assumption about the bound on the $G$ gradient dissimilarity can easily be extended to $(G, B)$ gradient dissimilarity used by (Karimireddy et al., 2019):

$$\mathbb{E}_i \|\nabla f_i(\boldsymbol{x})\|^2 \le G^2 + B^2 \|\nabla f(\boldsymbol{x})\|^2 \,. \tag{7}$$

All the proofs in the paper extend in a straightforward manner to the above weaker notion. Since this notion does not present any novel technical challenge, we omit it in the rest of the proofs. Note however that the above weaker notion can potentially capture the fact that by increasing the model capacity, we can reduce $G$. In the extreme case, by taking a sufficiently over-parameterized model, it is possible to make $G = 0$ in certain settings (Vaswani et al., 2018). However, this comes both at a cost of increased resource requirements (i.e. higher memory and compute requirements per step) but can also result in other constants increasing (e.g. $B$ and $L$).

The second crucial definition we use in this work is that of $\delta$ bounded *Hessian* dissimilarity (A2). This has been used previously in the analyses of distributed (Shamir et al., 2014; Arjevani and Shamir, 2015; Reddi et al., 2016) and federated learning (Karimireddy et al., 2020), but has been restricted to quadratics. Here, we show how to extend both the notion as well as the analysis to general smooth functions. The main manner we will use this assumption is in Lemma 3 to claim that for any $\boldsymbol{x}$ and $\boldsymbol{y}$ the following holds:

$$\mathbb{E}\|\nabla f_i(\boldsymbol{y};\zeta) - \nabla f_i(\boldsymbol{x};\zeta) + \nabla f(\boldsymbol{x}) - \nabla f(\boldsymbol{y})\|^2 \le \delta^2 \|\boldsymbol{y} - \boldsymbol{x}\|^2 \,. \tag{8}$$

Here the expectation is both over $\zeta$ as well as the choice of client $i$. To understand what the above condition means, it is illuminating to define $\Psi_i(\boldsymbol{z};\zeta) = f_i(\boldsymbol{z};\zeta) - f(\boldsymbol{z})$. Then, we can rewrite (A2) and (8) respectively as

$$\|\nabla^2 \Psi_i(\boldsymbol{z};\zeta)\| \le \delta \quad \text{and} \quad \mathbb{E}\|\nabla \Psi_i(\boldsymbol{y};\zeta) - \nabla \Psi_i(\boldsymbol{x};\zeta)\|^2 \le \delta^2 \|\boldsymbol{y} - \boldsymbol{x}\|^2 \,.$$

Thus (8) and (A2) are both different notions of smoothness of $\Psi_i(\boldsymbol{x};\zeta)$ (formal definition of smoothness will follow soon). The latter definition closely matches the notion of *squared-smoothness* used by Arjevani et al. (2019) and is a promising relaxation of (A2). However, we run into some technical issues since in our case the variable $\boldsymbol{y}$ can also be a random variable and depend on the choice of the client $i$. Extending our results to this weaker notion of Hessian-similarity and proving tight non-convex lower bounds is an exciting theoretical challenge.

Finally note that if the functions $f_i(\boldsymbol{x};\zeta)$ are assumed to be smooth as in (Shamir et al., 2014; Arjevani and Shamir, 2015; Karimireddy et al., 2020), then $\Psi_i((\boldsymbol{x};\zeta)$ is $2L$-smooth. Thus, we *always* have that $\delta \le 2L$. But, as Shamir et al. (2014) show, it is possible to have $\delta \ll L$ if the data distribution amongst the clients is similar. Further, the lower bound from Arjevani and Shamir (2015) proves that Hessian-similarity is the crucial quantity capturing the number of rounds of communication required for distributed/federated optimization.

We next define the terms smoothness and strong-convexity which we repeatedly use in the paper.

**(A3)** $f$ is **L-smooth** and satisfies:

$$\|\nabla f(\boldsymbol{x}) - \nabla f(\boldsymbol{y})\| \le L\|\boldsymbol{x} - \boldsymbol{y}\| \,, \text{ for any } \boldsymbol{x}, \boldsymbol{y} \,. \tag{9}$$

The assumption (A3) also implies the following quadratic upper bound on $f$

$$f(\boldsymbol{y}) \le f(\boldsymbol{x}) + \langle \nabla f(\boldsymbol{x}), \boldsymbol{y} - \boldsymbol{x} \rangle + \frac{L}{2}\|\boldsymbol{y} - \boldsymbol{x}\|^2 \,. \tag{10}$$

Further, if $f$ is twice-differentiable, (A3) implies that $\|\nabla^2 f(\boldsymbol{x})\| \le \beta$ for any $\boldsymbol{x}$.

**(A4)** $f$ is $\mu$-**PL strongly convex** (Karimi et al., 2016) for $\mu > 0$ if it satisfies:

$$\|\nabla f(\boldsymbol{x})\|^2 \ge 2\mu(f(\boldsymbol{x}) - f^\star) \,.$$

Note that PL-strong convexity is much weaker than the standard notion of strong-convexity (Karimi et al., 2016).

## D.2 Some technical lemmas

Now we cover some technical lemmas which are useful for computations later on. First, we state a relaxed triangle inequality true for the squared $\ell_2$ norm.

**Lemma 1** (relaxed triangle inequality). *Let $\{\boldsymbol{v}_1, \ldots, \boldsymbol{v}_\tau\}$ be $\tau$ vectors in $\mathbb{R}^d$. Then the following are true:*

1. $\|\boldsymbol{v}_i + \boldsymbol{v}_j\|^2 \leq (1+c)\|\boldsymbol{v}_i\|^2 + (1+\frac{1}{c})\|\boldsymbol{v}_j\|^2$ *for any $c > 0$, and*

2. $\|\sum_{i=1}^\tau \boldsymbol{v}_i\|^2 \leq \tau \sum_{i=1}^\tau \|\boldsymbol{v}_i\|^2.$

*Proof.* The proof of the first statement for any $c > 0$ follows from the identity:

$$\|\boldsymbol{v}_i + \boldsymbol{v}_j\|^2 = (1+c)\|\boldsymbol{v}_i\|^2 + (1+\tfrac{1}{c})\|\boldsymbol{v}_j\|^2 - \|\sqrt{c}\boldsymbol{v}_i + \tfrac{1}{\sqrt{c}}\boldsymbol{v}_j\|^2 \,.$$

For the second inequality, we use the convexity of $\boldsymbol{x} \to \|\boldsymbol{x}\|^2$ and Jensen's inequality

$$\left\|\frac{1}{\tau}\sum_{i=1}^\tau \boldsymbol{v}_i\right\|^2 \leq \frac{1}{\tau}\sum_{i=1}^\tau \|\boldsymbol{v}_i\|^2 \,. \qquad \square$$

Next we state an elementary lemma about expectations of norms of random vectors.

**Lemma 2** (separating mean and variance). *Let $\{\Xi_1, \ldots, \Xi_\tau\}$ be $\tau$ random variables in $\mathbb{R}^d$ which are not necessarily independent. First suppose that their mean is $\mathbb{E}[\Xi_i] = \xi_i$ and variance is bounded as $\mathbb{E}[\|\Xi_i - \xi_i\|^2] \leq \sigma^2$. Then, the following holds*

$$\mathbb{E}[\|\sum_{i=1}^\tau \Xi_i\|^2] \leq \|\sum_{i=1}^\tau \xi_i\|^2 + \tau^2\sigma^2 \,.$$

*Now instead suppose that their* conditional mean *is $\mathbb{E}[\Xi_i|\Xi_{i-1}, \ldots \Xi_1] = \xi_i$ i.e. the variables $\{\Xi_i - \xi_i\}$ form a martingale difference sequence, and the variance is bounded by $\mathbb{E}[\|\Xi_i - \xi_i\|^2] \leq \sigma^2$ as before. Then we can show the tighter bound*

$$\mathbb{E}[\|\sum_{i=1}^\tau \Xi_i\|^2] \leq 2\|\sum_{i=1}^\tau \xi_i\|^2 + 2\tau\sigma^2 \,.$$

*Proof.* For any random variable $X$, $\mathbb{E}[X^2] = (\mathbb{E}[X - \mathbb{E}[X]])^2 + (\mathbb{E}[X])^2$ implying

$$\mathbb{E}[\|\sum_{i=1}^\tau \Xi_i\|^2] = \|\sum_{i=1}^\tau \xi_i\|^2 + \mathbb{E}[\|\sum_{i=1}^\tau \Xi_i - \xi_i\|^2] \,.$$

Expanding the above expression using relaxed triangle inequality (Lemma 1) proves the first claim:

$$\mathbb{E}[\|\sum_{i=1}^\tau \Xi_i - \xi_i\|^2] \leq \tau \sum_{i=1}^\tau \mathbb{E}[\|\Xi_i - \xi_i\|^2] \leq \tau^2\sigma^2 \,.$$

For the second statement, $\xi_i$ is not deterministic and depends on $\Xi_{i-1}, \ldots, \Xi_1$. Hence we have to resort to the cruder relaxed triangle inequality to claim

$$\mathbb{E}[\|\sum_{i=1}^\tau \Xi_i\|^2] \leq 2\|\sum_{i=1}^\tau \xi_i\|^2 + 2\,\mathbb{E}[\|\sum_{i=1}^\tau \Xi_i - \xi_i\|^2]$$

and then use the tighter expansion of the second term:

$$\mathbb{E}[\|\sum_{i=1}^\tau \Xi_i - \xi_i\|^2] = \sum_{i,j} \mathbb{E}\big[(\Xi_i - \xi_i)^\top(\Xi_j - \xi_j)\big] = \sum_i \mathbb{E}\big[\|\Xi_i - \xi_i\|^2\big] \leq \tau\sigma^2 \,.$$

The cross terms in the above expression have zero mean since $\{\Xi_i - \xi_i\}$ form a martingale difference sequence. $\qquad \square$

# E  PROPERTIES OF FUNCTIONS WITH $\delta$ BOUNDED HESSIAN DISSIMILARITY

We now study two lemmas which hold for any functions which satisfy (A2). The first is closely related to the notion of smoothness (A3).

**Lemma 3** (similarity). *The following holds for any two functions $f_i(\cdot; \zeta)$ and $f(\cdot)$ satisfying* (A2), *and any $\boldsymbol{x}, \boldsymbol{y}$:*

$$\|\nabla f_i(\boldsymbol{y}; \zeta) - \nabla f_i(\boldsymbol{x}; \zeta) + \nabla f(\boldsymbol{x}) - \nabla f(\boldsymbol{y})\|^2 \le \delta^2 \|\boldsymbol{y} - \boldsymbol{x}\|^2 \,.$$

*Proof.* Consider the function $\Psi(\boldsymbol{z}) := f_i(\boldsymbol{z}; \zeta) - f(\boldsymbol{z})$. By the assumption (A2), we know that $\|\nabla^2 \Psi(\boldsymbol{z})\| \le \delta$ for all $\boldsymbol{z}$ i.e. $\Psi$ is $\delta$-smooth. By standard arguments based on taking limits (Nesterov, 2018), this implies that

$$\|\nabla \Psi(\boldsymbol{y}) - \nabla \Psi(\boldsymbol{x})\| \le \delta \|\boldsymbol{y} - \boldsymbol{x}\| \,.$$

Plugging back the definition of $\Psi$ into the above inequality proves the lemma. $\qquad\square$

Next, we see how weakly-convex functions satisfy a weaker notion of "averaging does not hurt".

**Lemma 4** (averaging). *Suppose $f$ is $\delta$-weakly convex. Then, for any $\gamma \ge \delta$, and a sequence of parameters $\{\boldsymbol{y}_i\}_{i \in \mathcal{S}}$ and $\boldsymbol{x}$:*

$$\frac{1}{|\mathcal{S}|} \sum_{i \in \mathcal{S}} f(\boldsymbol{y}_i) + \frac{\gamma}{2} \|\boldsymbol{x} - \boldsymbol{y}_i\|^2 \ge f(\bar{\boldsymbol{y}}) + \frac{\gamma}{2} \|\boldsymbol{x} - \bar{\boldsymbol{y}}\|^2 \,, \text{ where } \bar{\boldsymbol{y}} := \frac{1}{|\mathcal{S}|} \sum_{i \in \mathcal{S}} \boldsymbol{y}_i \,.$$

*Proof.* Since $f$ is $\delta$-weakly convex, $\Phi(\boldsymbol{z}) := f(\boldsymbol{z}) + \frac{\gamma}{2} \|\boldsymbol{z} - \boldsymbol{x}\|^2$ is convex. This proves the claim since $\frac{1}{|\mathcal{S}|} \sum_{i \in \mathcal{S}} \Phi(\boldsymbol{y}_i) \le \Phi(\bar{\boldsymbol{y}})$. $\qquad\square$

## F    ANALYSIS OF MIMESGD (WITHOUT MOMENTUM)

Let us rewrite the MimeSGD update using notation convenient for analysis. In each round $t$, we sample clients $\mathcal{S}^t$ such that $|\mathcal{S}^t| = S$. The server communicates the server parameters $\boldsymbol{x}^{t-1}$ as well as the average gradient across the sampled clients $\boldsymbol{c}^{t-1}$ defined as

$$\boldsymbol{c}^{t-1} = \frac{1}{S} \sum_{i \in \mathcal{S}^t} \nabla f_i(\boldsymbol{x}^{t-1}) \,. \tag{11}$$

Note that computing $\boldsymbol{c}^{t-1}$ itself requires two rounds of communication. But from a theoretical viewpoint, this only changes the communication rounds required by a constant factor and hence we ignore this issue. Practically, we recommend using MimeLite if this additional rounds of communication are an issue.

Then each client $i \in \mathcal{S}^t$ makes a copy $\boldsymbol{y}_{i,0}^t = \boldsymbol{x}^{t-1}$ and perform $K$ local client updates. In each local client update $k \in [K]$, the client samples a dataset $\zeta_{i,k}^t$ and

$$\boldsymbol{y}_{i,k}^t = \boldsymbol{y}_{i,k-1}^t - \eta(\nabla f_i(\boldsymbol{y}_{i,k-1}^t; \zeta_{i,k}^t) - \nabla f_i(\boldsymbol{x}^{t-1}; \zeta_{i,k}^t) + \boldsymbol{c}^{t-1}) \,. \tag{12}$$

After $K$ such local updates, the server then aggregates the new client parameters as

$$\boldsymbol{x}^t = \frac{1}{S} \sum_{i \in \mathcal{S}^t} \boldsymbol{y}_{i,K}^t \,. \tag{13}$$

**Variance of update.**    Consider the local update at step $k$ on client $i$, dropping superscript $t$

$$\boldsymbol{y}_{i,k} = \boldsymbol{y}_{i,k-1} - \eta \boldsymbol{d}_{i,k}, \text{ where } \boldsymbol{d}_{i,k} := \nabla f_i(\boldsymbol{y}_{i,k-1}; \zeta_{i,k}) - \nabla f_i(\boldsymbol{x}; \zeta_{i,k}) + \boldsymbol{c} \,.$$

**Lemma 5.** *Given that assumptions* (A1) *and* (A2) *are satisfied, each client update satisfies*

$$\mathbb{E}\|\boldsymbol{d}_{i,k}\|^2 \le \frac{3G^2}{S} + 3\delta^2\|\boldsymbol{y}_{i,k-1} - \boldsymbol{x}\|^2 + 3\|\nabla f(\boldsymbol{y}_{i,k-1})\|^2 \,.$$

*Proof.* Starting from the definition of $\boldsymbol{d}_{i,k}$ and the relaxed triangle inequality,

$$
\begin{aligned}
\|\boldsymbol{d}_{i,k}\|^2 &= \|\nabla f_i(\boldsymbol{y}_{i,k-1}; \zeta_{i,k}) - \nabla f_i(\boldsymbol{x}; \zeta_{i,k}) + \boldsymbol{c}\|^2 \\
&= \|\nabla f_i(\boldsymbol{y}_{i,k-1}; \zeta_{i,k}) - \nabla f_i(\boldsymbol{x}; \zeta_{i,k}) + \nabla f(\boldsymbol{x}) - \nabla f(\boldsymbol{y}_{i,k-1}) + (\boldsymbol{c} - \nabla f(\boldsymbol{x})) + \nabla f(\boldsymbol{y}_{i,k-1})\|^2 \\
&\le 3\|\nabla f_i(\boldsymbol{y}_{i,k-1}; \zeta_{i,k}) - \nabla f_i(\boldsymbol{x}; \zeta_{i,k}) + \nabla f(\boldsymbol{x}) - \nabla f(\boldsymbol{y}_{i,k-1})\|^2 + 3\|\boldsymbol{c} - \nabla f(\boldsymbol{x})\|^2 + 3\|\nabla f(\boldsymbol{y}_{i,k-1})\|^2 \\
&\le 3\delta^2\|\boldsymbol{y}_{i,k-1} - \boldsymbol{x}\|^2 + 3\|\boldsymbol{c} - \nabla f(\boldsymbol{x})\|^2 + 3\|\nabla f(\boldsymbol{y}_{i,k-1})\|^2 \,.
\end{aligned}
$$

We used Lemma 3 to bound the first term. Taking expectations on both sides to bound the second term via (A1) yields the lemma.                                                                                     □

**Distance moved in each round.**    We show that the distance moved by a client in each round during the $K$ updates can be controlled. To further reduce the burden of notation, we will drop he subscript $i, k$ and refer $\boldsymbol{y}_{i,k-1}$ simply as $\boldsymbol{y}$ and $\boldsymbol{y}_{i,k}$ as $\boldsymbol{y}^+$.

**Lemma 6.** *For update following* (12) *for* $\eta \le \frac{1}{4K\delta}$ *satisfying* (A1) *and* (A2)*, we have at any step $k$,*

$$\mathbb{E}\|\boldsymbol{y}^+ - \boldsymbol{x}\|^2 \le \left(1 + \tfrac{2}{K}\right)\|\boldsymbol{y} - \boldsymbol{x}\|^2 + 6K\eta^2\frac{G^2}{S} + 6K\eta^2\|\nabla f(\boldsymbol{y})\|^2 \,.$$

*Proof.* Starting from the update (12) and the relaxed triangle inequality Lemma 1 with $c = K \ge 1$,

$$
\begin{aligned}
\mathbb{E}\|\boldsymbol{y}^+ - \boldsymbol{x}\|^2 &= \mathbb{E}\|\boldsymbol{y} - \eta \boldsymbol{d} - \boldsymbol{x}\|^2 \\
&\le (1 + \tfrac{1}{c})\,\mathbb{E}\|\boldsymbol{y} - \boldsymbol{x}\|^2 + (1 + c)\eta^2\,\mathbb{E}\|\boldsymbol{d}\|^2 \\
&\le (1 + \tfrac{1}{K})\,\mathbb{E}\|\boldsymbol{y} - \boldsymbol{x}\|^2 + 3(1 + K)\eta^2\frac{G^2}{S} + 3(1 + K)\eta^2\delta^2\|\boldsymbol{y} - \boldsymbol{x}\|^2 + 3(1 + K)\eta^2\|\nabla f(\boldsymbol{y})\|^2 \\
&\le (1 + \tfrac{1}{K} + 6K\eta^2\delta^2)\,\mathbb{E}\|\boldsymbol{y} - \boldsymbol{x}\|^2 + 6K\eta^2\frac{G^2}{S} + 6K\eta^2\|\nabla f(\boldsymbol{y})\|^2 \,.
\end{aligned}
$$

The second to last step used the variance bound in Lemma 5. The proof now follows from the restriction on step-size since $16K^2\eta^2\delta^2 \le 1$.                                                          □

**Progress in one client update.** We now have the tools required to keep track of the progress made in one round.

**Lemma 7.** *For any constant* $\mu \geq 0$ *and each step of MimeSGD with step size* $\eta \leq \min\left(\frac{1}{18L}, \frac{1}{756\delta K}, \frac{1}{42\mu K}\right)$, *and given that* (A1)–(A3) *hold, we have*

$$\frac{\eta}{4}\,\mathbb{E}\|\nabla f(\boldsymbol{y}_{i,k-1}^t)\|^2 \leq A_{i,k-1}^t - A_{i,k}^t + \frac{(255KL\eta^2)G^2}{2S}\,,$$

*where we define*

$$A_{i,k}^t := \mathbb{E}[f(\boldsymbol{y}_{i,k}^t)] + \delta\left(1 + \tfrac{3}{K}\right)^{K-k}\mathbb{E}\|\boldsymbol{y}_{i,k}^t - \boldsymbol{x}^{t-1}\|^2\,, \text{ and}$$

$$A_{i,k-1}^t := \mathbb{E}[f(\boldsymbol{y}_{i,k-1}^t)] + \delta(1-\mu\eta)\left(1 + \tfrac{3}{K}\right)^{K-k+1}\mathbb{E}\|\boldsymbol{y}_{i,k-1}^t - \boldsymbol{x}^{t-1}\|^2\,.$$

*Proof.* The assumption that $f$ is $L$-smooth implies a quadratic upper bound (10). Using this in our case, we have

$$\mathbb{E}[f(\boldsymbol{y}^+)] - \mathbb{E}[f(\boldsymbol{y})] \leq -\eta\,\mathbb{E}[\langle\nabla f(\boldsymbol{y}), \boldsymbol{d}\rangle] + \frac{L\eta^2}{2}\,\mathbb{E}\|\boldsymbol{d}\|^2$$

$$= \underbrace{-\eta\,\mathbb{E}[\langle\nabla f(\boldsymbol{y}), \nabla f_i(\boldsymbol{y};\zeta) - \nabla f_i(\boldsymbol{x};\zeta) + \boldsymbol{c}\rangle]}_{\mathcal{T}_1} + \underbrace{\frac{L\eta^2}{2}\,\mathbb{E}\|\boldsymbol{d}\|^2}_{\mathcal{T}_2}\,.$$

Let us examine the terms $\mathcal{T}_1$ and $\mathcal{T}_2$ separately. By our variance bound Lemma 5, we have that

$$\mathcal{T}_2 \leq \frac{3L\eta^2 G^2}{2S} + \frac{3L\eta^2\delta^2}{2}\|\boldsymbol{y} - \boldsymbol{x}\|^2 + \frac{3L\eta^2}{2}\|\nabla f(\boldsymbol{y})\|^2\,.$$

To simplify $\mathcal{T}_1$, the biggest obstacle is that $\mathbb{E}[\nabla f_i(\boldsymbol{y};\zeta)] \neq \nabla f(\boldsymbol{y})$ since $\boldsymbol{y}$ itself depends on the sampling of the client $i$. Only the server gradient is unbiased and $\mathbb{E}[\boldsymbol{c}] = \nabla f(\boldsymbol{x})$. Instead we will use the similarity of the functions as in Lemma 3:

$$\mathcal{T}_1 = -\eta\,\mathbb{E}[\langle\nabla f(\boldsymbol{y}), \nabla f_i(\boldsymbol{y};\zeta) - \nabla f_i(\boldsymbol{x};\zeta) + \nabla f(\boldsymbol{x})\rangle]$$

$$\leq -\frac{\eta}{2}\,\mathbb{E}\|\nabla f(\boldsymbol{y})\|^2 + \frac{\eta}{2}\|\nabla f_i(\boldsymbol{y};\zeta) - \nabla f_i(\boldsymbol{x};\zeta) + \nabla f(\boldsymbol{x}) - \nabla f(\boldsymbol{y})\|^2$$

$$\leq -\frac{\eta}{2}\,\mathbb{E}\|\nabla f(\boldsymbol{y})\|^2 + \frac{\eta\delta^2}{2}\,\mathbb{E}\|\boldsymbol{y} - \boldsymbol{x}\|^2\,.$$

The first inequality above used that for any $a, b$, the following holds $-2ab = (a-b)^2 - a^2 - b^2 \leq (a-b)^2 - a^2$. The second used the similarity Lemma 3. Combining the terms $\mathcal{T}_1$ and $\mathcal{T}_2$ together, we have

$$\mathbb{E}[f(\boldsymbol{y}^+)] - \mathbb{E}[f(\boldsymbol{y})] \leq \frac{(3L\eta^2 - \eta)}{2}\,\mathbb{E}\|\nabla f(\boldsymbol{y})\|^2 + \frac{(\eta\delta^2 + 3L\eta^2\delta^2)}{2}\,\mathbb{E}\|\boldsymbol{y} - \boldsymbol{x}\|^2 + \frac{3L\eta^2 G^2}{2S}\,.$$

To bound the distance between $\boldsymbol{y}$ and $\boldsymbol{x}$, we use Lemma 6 multiplied on both sides by $\delta\left(1 + \tfrac{3}{K}\right)^{K-k}$. Note that $\delta \leq \delta\left(1 + \tfrac{3}{K}\right)^{K-k} \leq 21\delta$. This gives us for any constant $\mu \geq 0$

$$\delta\left(1 + \tfrac{3}{K}\right)^{K-k}\mathbb{E}\|\boldsymbol{y}^+ - \boldsymbol{x}\|^2 \leq \delta\left(1 + \tfrac{3}{K}\right)^{K-k}\left(1 + \tfrac{2}{K}\right)\|\boldsymbol{y} - \boldsymbol{x}\|^2 + 6K\delta\left(1 + \tfrac{3}{K}\right)^{K-k}\eta^2\frac{G^2}{S} +$$

$$6K\delta\left(1 + \tfrac{3}{K}\right)^{K-k}\eta^2\|\nabla f(\boldsymbol{y})\|^2$$

$$\leq \delta(1-\mu\eta)\left(1 + \tfrac{3}{K}\right)^{K-(k-1)}\|\boldsymbol{y} - \boldsymbol{x}\|^2 + 6K\delta\left(1 + \tfrac{3}{K}\right)^{K-k}\eta^2\frac{G^2}{S} +$$

$$6K\delta\left(1 + \tfrac{3}{K}\right)^{K-k}\eta^2\|\nabla f(\boldsymbol{y})\|^2 + \left(1 + \tfrac{3}{K}\right)^{K-k}(\mu\eta\delta - \tfrac{\delta}{K})\|\boldsymbol{y} - \boldsymbol{x}\|^2$$

$$\leq \delta\left(1 + \tfrac{3}{K}\right)^{K-(k-1)}\|\boldsymbol{y} - \boldsymbol{x}\|^2 + 126K\delta\eta^2\frac{G^2}{S} + 126K\delta\eta^2\|\nabla f(\boldsymbol{y})\|^2$$

$$+ (21\mu\eta\delta - \tfrac{\delta}{K})\|\boldsymbol{y} - \boldsymbol{x}\|^2\,.$$

The second inequality from the last used that $1+2/K < (1+3/K)(1-\mu\eta) = (1+2/K)+(1/K-(1+3/K)\mu\eta)$. This is true by our restriction that $\eta < \frac{1}{42\mu K}$, which implies $(1+3/K)\mu\eta < 4\mu\eta < 1/(10K)$ and so that $(1/K-(1+3/K)\mu\eta) > 0$. Adding the two bounds, we get the following recursion

$$\underbrace{\mathbb{E}[f(\boldsymbol{y}^+)] + \delta\left(1+\tfrac{3}{K}\right)^{K-k}\mathbb{E}\|\boldsymbol{y}^+ - \boldsymbol{x}\|^2}_{=:A_{i,k}} \leq \underbrace{\mathbb{E}[f(\boldsymbol{y})] + \delta\left(1+\tfrac{3}{K}\right)^{K-(k-1)}(1-\mu\eta)\|\boldsymbol{y}-\boldsymbol{x}\|^2}_{=:A_{i,k-1}}$$

$$+ \frac{(252K\delta\eta^2 + 3L\eta^2 - \eta)}{2}\mathbb{E}\|\nabla f(\boldsymbol{y})\|^2$$

$$+ \left(\frac{(\eta\delta^2 + 3L\eta^2\delta^2 + 42\mu\eta\delta)}{2} - \frac{\delta}{K}\right)\mathbb{E}\|\boldsymbol{y}-\boldsymbol{x}\|^2$$

$$+ \frac{(3L\eta^2 + 252K\delta\eta^2)G^2}{2S}$$

Now, note that our constraint on the step-size $\eta \leq \min(\frac{1}{18L}, \frac{1}{756\delta K})$ implies that $252K\delta\eta^2 + 3L\eta^2 \leq \frac{\eta}{2}$ and $K(\eta\delta^2 + 3L\eta^2\delta^2 + 42\mu\eta\delta) \leq 2\delta$. Plugging this into the above bound and recalling that $\delta \leq L$ finishes the proof. $\square$

**Convergence for PL strongly-convex functions.** We will unroll the one step progress Lemma 7 to compute a linear rate.

**Theorem II.** *Suppose that (A1)–(A4) are satisfied for $\mu > 0$. Then the updates of MimeSGD with step-size $\eta = \min(\eta_{\max}, \tilde{\mathcal{O}}\left(\frac{1}{\mu TK}\right))$ for $\eta_{\max} = \min\left(\frac{1}{18L}, \frac{1}{756\delta K}, \frac{1}{42\mu K}\right)$ satisfy*

$$\mathbb{E}\|\nabla f(\boldsymbol{x}^{out})\|^2 \leq \tilde{\mathcal{O}}\left(\frac{LG^2}{\mu TS} + \frac{F}{\eta_{\max}}\exp\left(-\frac{\mu}{18L + 756\delta K + 42\mu K}TK\right)\right)$$

*where we define $F := f(\boldsymbol{x}^0) - f^\star$, $\bar{\boldsymbol{y}}_k^t$ is chosen to be $\boldsymbol{y}_{i,k}^t$ for $i \in \mathcal{S}^t$ uniformly at random, and the output $\boldsymbol{x}^{out}$ to be $\bar{\boldsymbol{y}}_k^t$ with probability proportional to $(1-\frac{\eta\mu}{4})^{KT-kt}$.*

*Proof.* Note that by PL strong convexity (A4), we have

$$\frac{\eta}{4}\|\nabla f(\boldsymbol{y})\|^2 \leq \frac{\eta}{8}\|\nabla f(\boldsymbol{y})\|^2 + \frac{\eta\mu}{4}(f(\boldsymbol{y}) - f^\star).$$

Using this, we can tighten the one step progress Lemma 7 as

$$\frac{\eta}{8}\mathbb{E}\|\nabla f(\boldsymbol{y}_{i,k-1}^t)\|^2 \leq \underbrace{\left(1-\frac{\mu\eta}{4}\right)\mathbb{E}[f(\boldsymbol{y}_{i,k-1}^t) - f^\star] + \delta\left(1-\frac{\mu\eta}{4}\right)\left(1+\tfrac{3}{K}\right)^{K-k+1}\mathbb{E}\|\boldsymbol{y}_{i,k-1}^t - \boldsymbol{x}^{t-1}\|^2}_{=:\left(1-\frac{\mu\eta}{4}\right)\Phi_{i,k-1}^t}$$

$$\underbrace{- \mathbb{E}[f(\boldsymbol{y}_{i,k}^t) - f^\star] + \delta\left(1+\tfrac{3}{K}\right)^{K-k}\mathbb{E}\|\boldsymbol{y}_{i,k}^t - \boldsymbol{x}^{t-1}\|^2}_{=:\Phi_{i,k}^t} + \frac{(255KL\eta^2)G^2}{2S},$$

Now take a weighted sum over the steps $k$ using weights $(1-\frac{\eta\mu}{4})^{K-k}$

$$\frac{\eta}{8}\sum_{k\in[K]}(1-\tfrac{\eta\mu}{4})^{K-k}\mathbb{E}\|\nabla f(\boldsymbol{y}_{i,k-1}^t)\|^2 \leq \Phi_{i,0}^t - \left(1-\tfrac{\mu\eta}{4}\right)^K\Phi_{i,K}^t + \sum_{k\in[K]}(1-\tfrac{\eta\mu}{4})^{K-k}\frac{(255KL\eta^2)G^2}{2S}.$$

By the initialization $\boldsymbol{y}_{i,0}^t = \boldsymbol{x}^{t-1}$ and hence $\Phi_{i,0}^t = \mathbb{E}f(\boldsymbol{x}^{t-1}) - f^\star$ and further by the averaging Lemma 4, we have

$$\frac{1}{S}\sum_{i\in\mathcal{S}}\Phi_{i,K}^t \geq \mathbb{E}f(\boldsymbol{x}^t) - f^\star.$$

Hence, on averaging over the clients we get the one round progress lemma

$$\frac{\eta}{8S} \sum_{k \in [K]} \sum_{i \in \mathcal{S}^t} (1 - \tfrac{\eta\mu}{4})^{K-k} \, \mathbb{E}\|\nabla f(\boldsymbol{y}^t_{i,k-1})\|^2 \leq \mathbb{E}\, f(\boldsymbol{x}^{t-1}) - f^\star - \left(1 - \tfrac{\mu\eta}{4}\right)^K (\mathbb{E}\, f(\boldsymbol{x}^t) - f^\star)$$

$$+ \sum_{k \in [K]} (1 - \tfrac{\eta\mu}{4})^{K-k} \frac{(255KL\eta^2)G^2}{2S} \,.$$

Now further taking a weighted average over the rounds $t \in [T]$ with weights proportional to $(1 - \tfrac{\eta\mu}{4})^{tK}$ gives

$$\frac{\eta}{8S} \sum_{t \in [T]} \sum_{k \in [K]} \sum_{i \in \mathcal{S}^t} (1 - \tfrac{\eta\mu}{4})^{KT-kt} \, \mathbb{E}\|\nabla f(\boldsymbol{y}^t_{i,k-1})\|^2 \leq \mathbb{E}\, f(\boldsymbol{x}^0) - f^\star$$

$$+ \sum_{t \in [T]} \sum_{k \in [K]} (1 - \tfrac{\eta\mu}{4})^{KT-kt} \frac{(255KL\eta^2)G^2}{2S} \,.$$

Finally, choosing the right step size, similar to Lemma 23 of (Karimireddy et al., 2020) yields the desired rate. □

**Convergence for general functions.** We will unroll the one step progress Lemma 7 to compute a sublinear rate.

**Theorem III.** *Suppose that* (A1)–(A3) *are satisfied. Then the updates of MimeSGD with step-size* $\eta = \min\left(\eta_{\max}, \frac{\sqrt{FS}}{\sqrt{255K^2LG^2T}}\right)$ *for* $\eta_{\max} = \min\left(\frac{1}{18L}, \frac{1}{756\delta K}\right)$ *satisfy*

$$\frac{1}{KTS} \sum_{k \in [K]} \sum_{t \in [T]} \sum_{i \in \mathcal{S}^t} \mathbb{E}\|\nabla f(\boldsymbol{y}^t_{i,k-1})\|^2 \leq \mathcal{O}\left(\frac{G\sqrt{LF}}{\sqrt{TS}} + \frac{(L + \delta K)F}{TK}\right).$$

*where we define* $F := f(\boldsymbol{x}^0) - f^\star$.

*Proof.* By summing over the equations from Lemma 7 for all local steps in one round we obtain

$$\frac{\eta}{2} \sum_{k=1}^K \mathbb{E}\|\nabla f(\boldsymbol{y}^t_{i,k-1})\|^2 \leq A^t_{i,0} - A^t_{i,K} + \frac{255K^2L\eta^2G^2}{2S} \,.$$

By the initialization $\boldsymbol{y}^t_{i,0} = \boldsymbol{x}^{t-1}$, hence $A^t_{i,0} = A^t_{j,0} = \mathbb{E}[f(\boldsymbol{x}^{t-1})]$ for all $i, j \in \mathcal{S}^t$. Furthermore, by Lemma 4

$$\frac{1}{|\mathcal{S}^t|} \sum_{i \in \mathcal{S}_t} A^t_{i,K} \geq \mathbb{E}[f(\boldsymbol{x}^t)] + \delta\|\boldsymbol{x}^{t-1} - \boldsymbol{x}^t\|^2 \geq \mathbb{E}[f(\boldsymbol{x}^t)] = A^{t+1}_{i,0}$$

This means that we can keep unrolling over all rounds, obtaining

$$\frac{\eta}{2S} \sum_{t=1}^T \sum_{k=1}^K \sum_{i \in \mathcal{S}^t} \mathbb{E}\|\nabla f(\boldsymbol{y}^t_{i,k-1})\|^2 \leq A^1_{i,0} - A^T_{i,K} + \frac{255TK^2L\eta^2G^2}{2S} \,.$$

By noting $A^1_{i,0} - A^T_{i,K} = (f(\boldsymbol{x}^0) - f^\star) - (\mathbb{E}[f(\boldsymbol{x}^T) - f^\star) \leq F$ and the choice of the stepsize the theorem follows. □

# G  ANALYSIS OF MIMEMVR (WITH MOMENTUM BASED VARIANCE REDUCTION)

In this section we see how to use momentum based variance reduction (Cutkosky and Orabona, 2019) to reduce the variance of the updates and improve convergence. It should be noted that MVR does not exactly fit the MIME framework (BASEALG) since it requires computing gradients at two points on the same batch. However, it is straightforward to extend the idea of MIME to MVR as we will now do. We use MVR as a theoretical justification for why the usual momentum works well in practice. An interesting future direction would be to adapt the algorithm and analysis of (Cutkosky and Mehta, 2020), which does fit the framework of MIME.

**MimeMVR algorithm.**  Now, we formally describe the MimeMVR algorithm. In each round $t$, we sample clients $\mathcal{S}^t$ such that $|\mathcal{S}^t| = S$. The server communicates the server parameters $\boldsymbol{x}^{t-1}$, the momentum $\boldsymbol{m}^{t-1}$ and the average gradient across the sampled clients $\boldsymbol{c}^{t-1}$ defined as

$$\boldsymbol{c}^{t-1} = \frac{1}{S} \sum_{i \in \mathcal{S}^t} \nabla f_i(\boldsymbol{x}^{t-2}) \,. \tag{14}$$

Note that both $\boldsymbol{c}^{t-1}$ and $\boldsymbol{m}^{t-1}$ use gradients and parameters from previous rounds (different from the previous section).

Then each client $i \in \mathcal{S}^t$ makes a copy $\boldsymbol{y}_{i,0}^t = \boldsymbol{x}^{t-1}$ and perform $K$ local client updates. In each local client update $k \in [K]$, the client samples a dataset $\zeta_{i,k}^t$ and

$$\begin{aligned}
\boldsymbol{y}_{i,k}^t &= \boldsymbol{y}_{i,k-1}^t - \eta \boldsymbol{d}_{i,k}^t \,, \text{ where} \\
\boldsymbol{d}_{i,k}^t &= a(\nabla f_i(\boldsymbol{y}_{i,k-1}^t; \zeta_{i,k}^t) - \nabla f_i(\boldsymbol{x}^{t-1}; \zeta_{i,k}^t) + \boldsymbol{c}^{t-1}) + (1-a)\boldsymbol{m}^{t-1} \\
&\quad + (1-a)(\nabla f_i(\boldsymbol{y}_{i,k-1}^t; \zeta_{i,k}^t) - \nabla f_i(\boldsymbol{x}^{t-1}; \zeta_{i,k}^t)) \,.
\end{aligned} \tag{15}$$

After $K$ such local updates, the server then aggregates the new client parameters as

$$\boldsymbol{x}^t = \frac{1}{S} \sum_{j \in \mathcal{S}^t} \boldsymbol{y}_{j,K}^t \,. \tag{16}$$

The momentum term is updated at the end of the round for $a \geq 0$ as

$$\boldsymbol{m}^t = \underbrace{a(\tfrac{1}{S} \sum_{j \in \mathcal{S}^t} \nabla f_j(\boldsymbol{x}^{t-1})) + (1-a)\boldsymbol{m}^{t-1}}_{\text{SGDm}} + \underbrace{(1-a)(\tfrac{1}{S} \sum_{j \in \mathcal{S}^t} \nabla f_j(\boldsymbol{x}^{t-1}) - \nabla f_j(\boldsymbol{x}^{t-2}))}_{\text{correction}} \,. \tag{17}$$

As we can see, the momentum update of MVR can be broken down into the usual SGDm update, and a correction. Intuitively, this correction term is very small since $f_i$ is smooth and $\boldsymbol{x}^{t-1} \approx \boldsymbol{x}^{t-2}$. Another way of looking at the update (17) is to note that if all functions are identical i.e. $f_j = f_k$ for any $j, k$, then (17) just becomes the usual gradient descent. Thus MimeMVR tries to maintain an exponential moving average of only the variance terms, reducing its bias. We refer to (Cutkosky and Orabona, 2019) for more detailed explanation of MVR.

**Momentum variance bound.**  We compute the variance of the server momentum $\boldsymbol{m}^{t-1}$. Define the variance term $V^t = \boldsymbol{m}^t - \nabla f(\boldsymbol{x}^{t-1})$. Then its expected norm can be bounded as follows.

**Lemma 8.** *For the momentum update (17), given (A1) and (A2), the following holds for any $a \in [0, 1]$ and $V^t := \boldsymbol{m}^t - \nabla f(\boldsymbol{x}^{t-1})$*

$$\mathbb{E}\|V^t\|^2 \leq (1-a)\,\mathbb{E}\|V^{t-1}\|^2 + 2\delta^2\,\mathbb{E}\|\boldsymbol{x}^{t-1} - \boldsymbol{x}^{t-2}\|^2 + \frac{2a^2 G^2}{S} \,.$$

*Proof.* Starting from the momentum update (17),

$$V^t = (1-a)V^{t-1}$$

$$+ (1-a)\left(\frac{1}{S}\sum_{j\in\mathcal{S}^t}(\nabla f_j(\boldsymbol{x}^{t-1}) - \nabla f_j(\boldsymbol{x}^{t-2})) - \nabla f(\boldsymbol{x}^{t-1}) + \nabla f(\boldsymbol{x}^{t-2})\right)$$

$$+ a\left(\frac{1}{S}\sum_{j\in\mathcal{S}^t}(\nabla f_j(\boldsymbol{x}^{t-1}) - \nabla f(\boldsymbol{x}^{t-1}))\right).$$

Now, the term $V^{t-1}$ does not have any information from round $t$ and hence is statistically independent of the rest of the terms. Further, the rest of the terms have mean 0. Hence, we can separate out the zero mean noise terms from the $V^{t-1}$ following Lemma 2 and then the relaxed triangle inequality Lemma 1 to claim

$$\mathbb{E}\|V^t\|^2 \le (1-a)^2\,\mathbb{E}\|V^{t-1}\|^2$$

$$+ 2(1-a)^2\left\|\frac{1}{S}\sum_{j\in\mathcal{S}^t}(\nabla f_j(\boldsymbol{x}^{t-1}) - \nabla f_j(\boldsymbol{x}^{t-2})) - \nabla f(\boldsymbol{x}^{t-1}) + \nabla f(\boldsymbol{x}^{t-2})\right\|^2$$

$$+ 2a^2\left\|\frac{1}{S}\sum_{j\in\mathcal{S}^t}(\nabla f_j(\boldsymbol{x}^{t-1}) - \nabla f(\boldsymbol{x}^{t-1}))\right\|^2$$

$$\le (1-a)^2\,\mathbb{E}\|V^{t-1}\|^2 + 2(1-a)^2\delta^2\|\boldsymbol{x}^{t-1} - \boldsymbol{x}^{t-2}\|^2 + \frac{2a^2G^2}{S}.$$

The inequality used the Hessian similarity Lemma 3 to bound the second term and the heterogeneity bound (A1) to bound the last term. Finally, note that $(1-a)^2 \le (1-a) \le 1$ for $a \in [0,1]$. $\qquad\square$

**Update variance bound.** Now we examine the variance of our update in each local step $\boldsymbol{d}_{i,k}^t$.

**Lemma 9.** *For the client update (15), given (A1) and (A2), the following holds for any $a \in [0,1]$*

$$\mathbb{E}\|\boldsymbol{d}_{i,k}^t - \nabla f(\boldsymbol{y}_{i,k-1}^t)\|^2 \le 3\,\mathbb{E}\|V^{t-1}\|^2 + 3\delta^2\,\mathbb{E}\|\boldsymbol{y}_{i,k-1}^t - \boldsymbol{x}^{t-2}\|^2 + \frac{3a^2G^2}{S}.$$

*Proof.* Starting from the client update (15), we can rewrite it as

$$\boldsymbol{d}_{i,k}^t - \nabla f(\boldsymbol{y}_{i,k-1}^t) = (1-a)V^{t-1}$$

$$+ \left(\nabla f_i(\boldsymbol{y}_{i,k-1}^t; \zeta_{i,k}^t) - \nabla f_i(\boldsymbol{x}^{t-2}; \zeta_{i,k}^t)\right) - \nabla f(\boldsymbol{y}_{i,k-1}^t + \nabla f(\boldsymbol{x}^{t-2}))$$

$$+ a\left(\frac{1}{S}\sum_{j\in\mathcal{S}^t}\nabla f_j(\boldsymbol{x}^{t-2}) - \nabla f(\boldsymbol{x}^{t-2})\right).$$

We can use the relaxed triangle inequality Lemma 1 to claim

$$\mathbb{E}\|\boldsymbol{d}_{i,k}^t - \nabla f(\boldsymbol{y}_{i,k-1}^t)\|^2 = 3(1-a)^2\,\mathbb{E}\|V^{t-1}\|^2$$

$$+ 3(1-a)^2\left\|(\nabla f_i(\boldsymbol{y}_{i,k-1}^t; \zeta_{i,k}^t) - \nabla f_i(\boldsymbol{x}^{t-2}; \zeta_{i,k}^t)) - (\nabla f(\boldsymbol{y}_{i,k-1}^t) - \nabla f(\boldsymbol{x}^{t-2}))\right\|^2$$

$$+ 3a^2\left\|\frac{1}{S}\sum_{j\in\mathcal{S}^t}\nabla f_j(\boldsymbol{x}^{t-2}) - \nabla f(\boldsymbol{x}^{t-2})\right\|^2$$

$$\le 3\,\mathbb{E}\|V^{t-1}\|^2 + 3\delta^2\|\boldsymbol{y}_{i,k-1}^t - \boldsymbol{x}^{t-2}\|^2 + \frac{3a^2G^2}{S}.$$

The last inequality used the Hessian similarity Lemma 3 to bound the second term and the heterogeneity bound (A1) to bound the last term. Also, $(1-a)^2 \le 1$ since $a \in [0,1]$. $\qquad\square$

**Distance moved in each step.** We show that the distance moved by a client in each step during the client update can be controlled.

**Lemma 10.** *For MimeMVR updates* (15) *with* $\eta \leq \frac{1}{6K\delta}$ *and given* (A1) *and* (A2), *the following holds*

$$\Delta_{i,k}^t \leq \left(1 + \frac{1}{K}\right)\Delta_{i,k-1}^t + 18\eta^2 K a^2 \frac{G^2}{S} + 18\eta^2 K \, \mathbb{E}\|V^{t-1}\|^2 + 6\eta^2 K \|\nabla f(\boldsymbol{y}_{i,k-1}^t)\|^2 \, ,$$

*where we define* $\Delta_{i,k}^t := \max\left(\mathbb{E}\|\boldsymbol{y}_{i,k}^t - \boldsymbol{x}^{t-2}\|^2 , \mathbb{E}\|\boldsymbol{y}_{i,k}^t - \boldsymbol{x}^{t-1}\|^2, \mathbb{E}\|\boldsymbol{x}^{t-1} - \boldsymbol{x}^{t-2}\|^2\right).$

*Proof.* Starting from the MimeMVR update (15) and the relaxed triangle inequality with $c = 2K$,

$$\mathbb{E}\|\boldsymbol{y}_{i,k}^t - \boldsymbol{x}^{t-2}\|^2 = \mathbb{E}\|\boldsymbol{y}_{i,k-1}^t - \eta\boldsymbol{d}_{i,k}^t - \boldsymbol{x}^{t-2}\|^2$$

$$\leq \left(1 + \frac{1}{2K}\right)\mathbb{E}\|\boldsymbol{y}_{i,k-1}^t - \boldsymbol{x}^{t-2}\|^2 + (2K+1)\eta^2 \, \mathbb{E}\|\boldsymbol{d}_{i,k}^t\|^2$$

$$\leq \left(1 + \frac{1}{2K}\right)\mathbb{E}\|\boldsymbol{y}_{i,k-1}^t - \boldsymbol{x}^{t-2}\|^2 + 6K\eta^2 \, \mathbb{E}\|\boldsymbol{d}_{i,k}^t - \nabla f(\boldsymbol{y}_{i,k-1}^t)\|^2$$

$$+ 6K\eta^2 \, \mathbb{E}\|\nabla f(\boldsymbol{y}_{i,k-1}^t)\|^2$$

$$\leq \left(1 + \frac{1}{2K} + 18K\eta^2\delta^2\right)\mathbb{E}\|\boldsymbol{y}_{i,k-1}^t - \boldsymbol{x}^{t-2}\|^2$$

$$+ 18K\eta^2 \, \mathbb{E}\|V^{t-1}\|^2 + \frac{18K\eta^2 a^2 G^2}{S} + 6K\eta^2 \, \mathbb{E}\|\nabla f(\boldsymbol{y}_{i,k-1}^t)\|^2 \, .$$

The last inequality used the update variance bound Lemma 9. We can simplify the expression further since $\eta \leq \frac{1}{6K\delta}$ implies $18K\eta^2\delta^2 \leq \frac{1}{2K}$. Similar computations for $\mathbb{E}\|\boldsymbol{y}_{i,k}^t - \boldsymbol{x}^{t-1}\|^2$ yield the lemma. $\qquad\square$

**Progress in one step.** Now we have all the tools required to compute the progress made in each round.

**Lemma 11.** *For any step of MimeMVR with step size* $\eta \leq \min\left(\frac{1}{L}, \frac{1}{40\delta K}\right)$ *and momentum parameter* $a = 1536\eta^2\delta^2 K^2$. *Then, given that* (A1)–(A3) *hold, we have*

$$\mathbb{E}[f(\boldsymbol{y}_{i,k}^t)] + \frac{3\eta}{a}\, \mathbb{E}\|V^t\|^2 + \frac{8\eta\delta^2 K}{a}\left(1 + \frac{2}{K}\right)^{K-k}\Delta_{i,k}^t$$

$$\leq \mathbb{E}[f(\boldsymbol{y}_{i,k-1}^t)] + \frac{3\eta}{a}\, \mathbb{E}\|V^{t-1}\|^2 + \frac{8\eta\delta^2 K}{a}\left(1 + \frac{2}{K}\right)^{K-(k-1)}\Delta_{i,k-1}^t$$

$$- \frac{\eta}{4}\, \mathbb{E}\|\nabla f(\boldsymbol{y}_{i,k-1}^t)\|^2 + \frac{11136\eta^3\delta^2 K^2 G^2}{S} \, .$$

*Proof.* The assumption that $f$ is $L$-smooth implies a quadratic upper bound (10).

$$f(\boldsymbol{y}_{i,k}^t) - f(\boldsymbol{y}_{i,k-1}^t) \leq -\eta\langle\nabla f(\boldsymbol{y}_{i,k-1}^t), \boldsymbol{d}_{i,k}^t\rangle + \frac{L\eta^2}{2}\|\boldsymbol{d}_{i,k}^t\|^2$$

$$= -\frac{\eta}{2}\|\nabla f(\boldsymbol{y}_{i,k-1}^t)\|^2 + \frac{L\eta^2 - \eta}{2}\|\boldsymbol{d}_{i,k}^t\|^2 + \frac{\eta}{2}\|\boldsymbol{d}_{i,k}^t - \nabla f(\boldsymbol{y}_{i,k-1}^t)\|^2 \, .$$

The second equality used the fact that for any $a, b$, $-2ab = (a-b)^2 - a^2 - b^2$. The second term can be removed since $\eta \leq \frac{1}{L}$. Taking expectation on both sides and using the update variance bound Lemma 9,

$$\mathbb{E}\, f(\boldsymbol{y}_{i,k}^t) - \mathbb{E}\, f(\boldsymbol{y}_{i,k-1}^t) \leq -\frac{\eta}{2}\, \mathbb{E}\|\nabla f(\boldsymbol{y}_{i,k-1}^t)\|^2 + \frac{3\eta a^2 G^2}{2S}$$

$$+ \frac{3\eta}{2}\, \mathbb{E}\|V^{t-1}\|^2 + \frac{3\eta\delta^2}{2}\, \mathbb{E}\|\boldsymbol{y}_{i,k-1}^t - \boldsymbol{x}^{t-2}\|^2$$

Multiplying the momentum variance bound Lemma 8 by $\frac{3\eta}{a}$, we have

$$\frac{3\eta}{a}\,\mathbb{E}\|V^t\|^2 \le \frac{3\eta}{a}\,\mathbb{E}\|V^{t-1}\|^2 + \frac{6\eta\delta^2}{a}\,\mathbb{E}\|\boldsymbol{x}^{t-1}-\boldsymbol{x}^{t-2}\|^2 + \frac{6\eta a G^2}{S} - 3\eta\,\mathbb{E}\|V^{t-1}\|^2\,.$$

We will also multiply the distance bound Lemma 10 by $\frac{8\eta\delta^2 K}{a}\left(1+\frac{2}{K}\right)^{K-k}$. Note that for any $K \ge 1$ and $k \in [K]$, we have $1 \le \left(1+\frac{2}{K}\right)^{K-k} \le 8$. Then we get

$$\frac{8\eta\delta^2 K}{a}\left(1+\frac{2}{K}\right)^{K-k}\Delta_{i,k}^t \le \frac{8\eta\delta^2 K}{a}\left(1+\frac{2}{K}\right)^{K-(k-1)}\Delta_{i,k-1}^t - \frac{8\eta\delta^2}{a}\Delta_{i,k-1}^t$$

$$+ 1152\eta^3\delta^2 K^2 a\frac{G^2}{S} + \frac{1152\eta^3\delta^2 K^2}{a}\,\mathbb{E}\|V^{t-1}\|^2 + \frac{384\eta^3\delta^2 K^2}{a}\|\nabla f(\boldsymbol{y}_{i,k-1}^t)\|^2\,,$$

where recall that we defined $\Delta_{i,k}^t := \max\Big(\mathbb{E}\|\boldsymbol{y}_{i,k}^t - \boldsymbol{x}^{t-2}\|^2, \mathbb{E}\|\boldsymbol{y}_{i,k}^t - \boldsymbol{x}^{t-1}\|^2, \mathbb{E}\|\boldsymbol{x}^{t-1}-\boldsymbol{x}^{t-2}\|^2\Big)$.
Combining the three inequalities together, we get

$$\mathbb{E}\,f(\boldsymbol{y}_{i,k}^t) + \frac{3\eta}{a}\,\mathbb{E}\|V^t\|^2 + \frac{8\eta\delta^2 K}{a}\left(1+\frac{2}{K}\right)^{K-k}\Delta_{i,k}^t$$

$$\le \mathbb{E}\,f(\boldsymbol{y}_{i,k-1}^t) + \frac{3\eta}{a}\,\mathbb{E}\|V^{t-1}\|^2 + \frac{8\eta\delta^2 K}{a}\left(1+\frac{2}{K}\right)^{K-(k-1)}\Delta_{i,k-1}^t$$

$$+ \left(\frac{384\eta^3\delta^2 K^2}{a} - \frac{\eta}{2}\right)\mathbb{E}\|\nabla f(\boldsymbol{y}_{i,k-1}^t)\|^2$$

$$+ \left(1152\eta^2\delta^2 K^2 + 6 + \frac{3a}{2}\right)\frac{a\eta G^2}{S}$$

$$+ \left(\frac{3\eta}{2} + \frac{1152\eta^3\delta^2 K^2}{a} - 3\eta\right)\mathbb{E}\|V^{t-1}\|^2$$

$$+ \left(\frac{3}{2} + \frac{6}{a} - \frac{8}{a}\right)\eta\delta^2\Delta_{i,k-1}^t\,.$$

Note that $\frac{1152\eta^3\delta^2 K^2}{a} = \frac{3\eta}{4}$ since we defined $a = 1536\eta^2\delta^2 K^2$. Further, $a \le 1$ when defined this way since we assumed $\eta \le \frac{1}{40\delta K}$. Similarly, the definition of $a$ implies that $\frac{384\eta^3\delta^2 K^2}{a} = \frac{\eta}{4}$. Thus, we can simplify the above expression as

$$\mathbb{E}\,f(\boldsymbol{y}_{i,k}^t) + \frac{3\eta}{a}\,\mathbb{E}\|V^t\|^2 + \frac{8\eta\delta^2 K}{a}\left(1+\frac{2}{K}\right)^{K-k}\Delta_{i,k}^t$$

$$\le \mathbb{E}\,f(\boldsymbol{y}_{i,k-1}^t) + \frac{3\eta}{a}\,\mathbb{E}\|V^{t-1}\|^2 + \frac{8\eta\delta^2 K}{a}\left(1+\frac{2}{K}\right)^{K-(k-1)}\Delta_{i,k-1}^t$$

$$- \frac{\eta}{4}\,\mathbb{E}\|\nabla f(\boldsymbol{y}_{i,k-1}^t)\|^2 + \frac{11136\eta^3\delta^2 K^2 G^2}{S}\,.$$

This proves the lemma. $\qquad\square$

**Progress in one round.** Let us sum over all the steps within a round to compute the progress made in a full round.

**Lemma 12.** *For any round of MimeMVR with step size $\eta \le \min\left(\frac{1}{L}, \frac{1}{40\delta K}\right)$ and momentum parameter $a = 1536\eta^2\delta^2 K^2$. Then, given that (A1)–(A3) hold, we have*

$$\frac{\eta}{4KS}\sum_{k\in[K], j\in\mathcal{S}^t}\mathbb{E}\|\nabla f(\boldsymbol{y}_{i,k-1}^t)\|^2 \le \Phi^{t-1} - \Phi^t + \frac{11136\eta^3\delta^2 K^2 G^2}{S}\,,$$

*where we define the sequence*

$$\Phi^t := \tfrac{1}{K}\,\mathbb{E}[f(\boldsymbol{x}^t)] + \frac{3\eta K}{a}\,\mathbb{E}\|V^t\|^2 + \frac{8\eta\delta^2}{a}\,\mathbb{E}\|\boldsymbol{x}^t - \boldsymbol{x}^{t-1}\|^2\,.$$

*Proof.* We start by summing Lemma 11 over the client updates

$$\frac{\eta}{4} \sum_{k \in [K]} \mathbb{E}\|\nabla f(\boldsymbol{y}_{i,k-1}^t)\|^2 \leq \mathbb{E}[f(\boldsymbol{y}_{i,0}^t)] + \frac{3\eta K}{a} \mathbb{E}\|V^{t-1}\|^2 + \frac{8\eta\delta^2 K}{a}\left(1 + \frac{2}{K}\right)^K \Delta_{i,0}^t$$

$$- \mathbb{E}[f(\boldsymbol{y}_{i,K}^t)] - \frac{3\eta K}{a} \mathbb{E}\|V^t\|^2 + \frac{8\eta\delta^2 K}{a}\Delta_{i,K}^t$$

$$+ \frac{11136\eta^3\delta^2 K^3 G^2}{S}.$$

Recall that we defined $\Delta_{i,k}^t := \max\left(\mathbb{E}\|\boldsymbol{y}_{i,k}^t - \boldsymbol{x}^{t-2}\|^2, \mathbb{E}\|\boldsymbol{y}_{i,k}^t - \boldsymbol{x}^{t-1}\|^2, \mathbb{E}\|\boldsymbol{x}^{t-1} - \boldsymbol{x}^{t-2}\|^2\right)$. Because $\boldsymbol{y}_{i,0}^t = \boldsymbol{x}^{t-1}$, we can simplify

$$\mathbb{E}[f(\boldsymbol{y}_{i,0}^t)] + \frac{8\eta\delta^2 K}{a}\Delta_{i,0}^t \leq \mathbb{E}[f(\boldsymbol{x}^{t-1})] + \frac{8\eta\delta^2 K}{a} \mathbb{E}\|\boldsymbol{x}^{t-1} - \boldsymbol{x}^{t-2}\|^2$$

Then by the averaging Lemma 4, we have

$$\frac{1}{S} \sum_{j \in \mathcal{S}^t} \mathbb{E}[f(\boldsymbol{y}_{j,K}^t)] + \frac{8\eta\delta^2 K}{a}\Delta_{j,K}^t \geq \frac{1}{S} \sum_{j \in \mathcal{S}} \mathbb{E}[f(\boldsymbol{y}_{j,K}^t)] + \mathbb{E}\|\boldsymbol{x}^{t-1} - \boldsymbol{y}_{j,K}^t\|^2$$

$$\geq \mathbb{E}[f(\boldsymbol{x}^t)] + \frac{8\eta\delta^2 K}{a} \mathbb{E}\|\boldsymbol{x}^{t-1} - \boldsymbol{x}^t\|^2.$$

So by averaging our recursion over the sampled clients, and diving our summation over the updates by $K$, we get

$$\frac{\eta}{4KS} \sum_{k \in [K], j \in \mathcal{S}^t} \mathbb{E}\|\nabla f(\boldsymbol{y}_{i,k-1}^t)\|^2 \leq \underbrace{\frac{1}{K} \mathbb{E}[f(\boldsymbol{x}^{t-1})] + \frac{3\eta}{a} \mathbb{E}\|V^{t-1}\|^2 + \frac{8\eta\delta^2}{a} \mathbb{E}\|\boldsymbol{x}^{t-1} - \boldsymbol{x}^{t-2}\|^2}_{=\Phi^{t-1}}$$

$$\underbrace{- \frac{1}{K} \mathbb{E}[f(\boldsymbol{x}^t)] + \frac{3\eta K}{a} \mathbb{E}\|V^t\|^2 + \frac{8\eta\delta^2}{a} \mathbb{E}\|\boldsymbol{x}^t - \boldsymbol{x}^{t-1}\|^2}_{=\Phi^t}$$

$$+ \frac{11136\eta^3\delta^2 K^2 G^2}{S}.$$

$\square$

**Theorem IV** (non-convex convergence of MimeMVR). *Let us run MimeMVR with step size $\eta \leq \min\left(\frac{1}{15K}\left(\frac{FS}{T\delta^2 G^2}\right)^{1/3}, \frac{1}{L}, \frac{1}{40\delta K}\right)$ and momentum parameter $a = 1536\eta^2\delta^2 K^2$. Then, given that* (A1)–(A3) *hold, we have*

$$\frac{1}{KST} \sum_{t \in [T]} \sum_{k \in [K]} \sum_{j \in \mathcal{S}^t} \mathbb{E}\|\nabla f(\boldsymbol{y}_{i,k-1}^t)\|^2 \leq \mathcal{O}\left(\left(\frac{(1+\delta)G^2 F}{ST}\right)^{2/3} + \frac{(L+\delta K)F}{KT}\right),$$

*where we define $F := f(\boldsymbol{x}^0) - f^\star$.*

*Proof.* Unroll the one round progress Lemma 12 and average over $T$ rounds to get

$$\frac{1}{KST} \sum_{t \in [T]} \sum_{k \in [K]} \sum_{j \in \mathcal{S}^t} \mathbb{E}\|f(\boldsymbol{y}_{i,k-1}^t)\|^2 \leq \frac{4(\Phi^0 - \Phi^T)}{\eta KT} + \frac{11136\eta^2\delta^2 K^2 G^2}{S}$$

$$\leq \frac{4(f(\boldsymbol{x}^0) - f^\star)}{\eta KT} + \frac{11136\eta^2\delta^2 K^2 G^2}{S}.$$

Our choice of step size now yields the desired rate.

$\square$

