# OpenReview forum: "Mime: Mimicking Centralized Stochastic Algorithms in Federated Learning"
_ICLR.cc/2021/Conference — Reject_

### Official Review · AnonReviewer2 · 2020-10-27
**This paper proposes a way to apply various variance reduction/momentum based method to the federated learning scenario, especially when there is distribution drift among the clients.**

**Rating:** 4
**Confidence:** 4

**Review:**

This paper proposes a way to apply various variance reduction/momentum based method to the federated learning scenario, especially when there is distribution drift among the clients. The main claim of this paper is that the global statistics (momentum, control variance et al) should be update at the server side only, which is helpful to reduce the bias of these terms. The paper also provides convergence analysis for their methods and attains the best convergence result for their MimeMVR method.
However, I find a key wrong in theoretical analysis. Specifically, in the proof of Lemma 7, the last second inequality in page 21 is wrong.  Thus I have difficulty judging the contribution of this paper.

Some comments are given as follows:

1.	In the proof of Lemma 7, when the authors bound the distance between y and x, how do you get at the last second inequality in page 21, it seems to rely on: 1 + 2/k < 1 + 2/k - mu*eta* 3/k, which obviously not hold.
2.	For the MimeMVR, the update rule for d^t_{i,k} in the equality (14), there is a  -1 term, which looks strange.  Why does this term exit?
3.	In the proof of Lemma 8, the second equality at page 25 should be inequality.
4.	In the proof of Lemma 9, there are typos using the brackets in the third equality at page 25.


---------------------------------------------------------------------------------------------------
After Rebuttals
---------------------------------------------------------------------------------------------------
Thanks for your responses. I still have doubts about the proof of lemma 7. You not only change from '2' to '3', you also add a new term (the fourth term). I admit that $1 + 2/k < (1 + 3/k)(1 – \mu*\eta)$ holds under your assumption, but you also need: $\mu*\eta> 1/k$, i.e. let the fourth term be greater than zero, to make the inequality hold, however, this contradicts the previous assumption. So I still believe this inequality does not hold.

---

> ### Author Response · Authors · 2020-11-14
> **We believe Lemma 7 is correct.**
>
> We thank the reviewer for their careful reading of our paper and proofs! The main concern raised seems to be the proof of Lemma 7. We believe this was a slight misunderstanding (mistaking a 2 for a 3). The inequality actually uses
> $$1 + 2/k < (1 + 3/k)(1 - \mu\eta) = (1 + 2/k) + (1/k - (1 + 3/k)\mu\eta).$$
> This inequality is true by our restriction that $\eta < \frac{1}{42 \mu k}$, and hence we have
> $$ (1 + 3/k)\mu\eta < 4\mu\eta < 1/(10k).$$
>
> The -1 term was an incorrect latex bracketing---it should read  c^{t-1} instead of c^t-1.
>
> We made a pass through our proofs again to weed out additional typos and also expanded upon the proof in Lemma 7 to explain the steps involved better. We hope that this convinces the reviewer of the correctness, and if so request them to kindly reevaluate our paper.

---

### Official Review · AnonReviewer3 · 2020-10-28
**Reviews**

**Rating:** 5
**Confidence:** 3

**Review:**

The paper proposes a new framework for solving federated learning. The authors consider a specific setting that there are many clients, and each client is allowed to compute the full gradient. The authors claim that the current setting’s main issue is the client drift, and the proposed framework can reduce such an issue and thus achieve a faster convergence rate. Here are my main concerns of the current paper:
1. The client drift issue is not clearly defined in this paper.
2. The comparisons in Table 1 are not fair since different methods have different assumptions. For example, the authors claim that the proposed MimeSGD algorithm can reduce communication costs compared with FedSGD. However, MimeSGD depends on the extra Hessian assumptions (A2).
3. Assumption A2 seems to be a very strong assumption. Intuitively, it contains two variances: one from the gradient dissimilarity and another from the within-client variance. Therefore, why can we expect that delta is very small? In addition, is the improvement of your framework comes from this assumption?
4. It seems that the algorithm with theoretical guarantees is different from the one implemented in experiments. To validate your results, the experiments should use the same algorithm.
5. The parameter beta in Theorem 1 seems to be a very small value. However, the experiments suggest a very large beta. This seems to be inconsistent with your theorem. Is this due to the different algorithms you implement in experiments?
6. What is the definition of x^out?
7. The proposed framework requires additional communications due to the correction term c. Did you consider this extra communication cost in your experiments?
8. In the experiments, since each client performs several epochs of updates, the comparison with the server-only method is not fair since it only performs one update each communication.
9. In Figure 2, why is there a spike for your Mime method?
10. Typos: relative slow-> relative low. vour-> our.

---

> ### Author Response · Authors · 2020-11-14
> **Subtle misunderstandings in definitions, assumptions, and setup.**
>
> We thank the reviewer for their careful reading of our work, and address concerns raised below. We believe we have addressed all of the issues raised, and if so kindly request the reviewer to re-evaluate our work appropriately. Due to character limit, we first address what we see as major concerns and the rest in a second comment.
>
> > MimeSGD uses extra Hessian assumptions (A2), which is unreasonably strong.
>
> Hessian distance is indeed a standard notion in distributed optimization and is known to tightly control the rounds of communication required (see [Arjevani & Shamir 2015]). Assumption (A2) is in fact **weaker** than assuming that the stochastic functions $f_i(x; \zeta)$ are smooth, the latter of which is common in FL literature (cf. Karimireddy et al. [2020]). If the stochastic functions are $L$-smooth, then (A2) is automatically satisfied for $\delta < 2L$. Hence (A2) is neither an extra assumption, nor is it a strong one. We added a more detailed discussion in Section D.1.
>
> > Is the improved result a consequence of (A2)?
>
> MimeMVR has two sources of improvements. In the first stochastic term we go from $O(1/\epsilon^2)$ of FedAvg to $O(1/\epsilon^{3/2})$ using momentum. Secondly, we show an additional improvement where we replace $L$ in the second order term with $\delta$. This comes with a more careful analysis of the bias terms which shows that the *weaker* (A2)  is sufficient, in place of the stronger smoothness assumptions.
>
> > The algorithm used experimentally is different from the one for which theory is proved.
>
> The difficulty of doing useful theory in deep learning is that it is still an open problem to formalize exactly when and why practically excellent algorithms (e.g. Adam, LARS, etc.) perform as well as they do. So adapting these schemes to the federated case in a principled manner would be tricky. We attempt to side step this issue as follows---we devise a generic framework (Mime) such that when combined with the algorithm with the best guarantees in the centralized setting (MVR), the resulting federated algorithm (MimeMVR) is also optimal. Note MVR is empirically outperformed by SGDm/Adam (cf. Cutkosky and Orabona [2019]) in the centralized setting. So we do not expect MimeMVR to achieve SOTA empirical performance. The hope instead is that the resulting framework Mime which preserves the good properties of MVR, also preserves the good properties of SOTA empirical algorithms such as SGDm, Adam, LARS, etc. This explains our choice of base algorithms---MVR for doing theory, and SGDm/Adam for experiments. As expected, MimeMVR indeed has the best theoretical performance, whereas MimeSGDm/MimeAdam have the best empirical performance.
>
> > Theorem 1 suggests using a very small $\beta$, inconsistent with the experiments.
>
> As we explained above, we do not expect Theorem 1 to exactly capture the behavior of empirical performance in deep learning. Having said that, The $\beta$ from theory is actually very large---$(1 - O(TG^2)^{-2/3})$ which asymptotically tends to 1 as T increases. Further, as the clients become more heterogeneous ($G$ increases), there is stronger client-drift and hence we need more momentum to compensate.
>
> > Experimental comparison is unfair since we use additional epochs/ need additional communication.
>
> In FL we only care about the number of **communication rounds**, and the bandwidth overhead due to sending twice the number of bits per round or the computational overhead due to performing multiple epochs is ignored [McMahan et al. 2017]. This is justified because most of the time in cross-device FL is spent in establishing connections with devices rather [Bonawitz et al. 2019]. If bandwidth is a concern, combining communication compression strategies e.g. [Suresh et al. 2017, Alistarh et al. 2017] with our methods is an interesting future work. We added a further discussion of this in Appendix A.2.
>
> References
> - Zhao et al. Federated learning with non-iid data. 2018
> - Karimireddy, et al. Scaffold: Stochastic controlled averaging for on-device federated learning. ICML 2020.
> - Charles and Konečný.  On the outsized importance of learning rates in local update methods. 2020.
> - Cutkosky and Orabona. Momentum-based variance reduction in non-convex SGD. NeurIPS 2019.
> - McMahanet al. Communication-efficient learning of deep networks from decentralized data. AISTATS 2017.
> - Bonawitz et al. Towards federated learning at scale: System design. 2019
> - Suresh et al. Distributed mean estimation with limited communication. ICML 2017.
> - Alistarh, Dan, et al. QSGD: Communication-efficient SGD via gradient quantization and encoding. NeurIPS 2017.
> - Su et al. A differential equation for modeling Nesterov’s accelerated gradient method: Theory and insights. NeurIPS 2014.

---

> > ### Author Response · Authors · 2020-11-14
> > **Addressing other minor points**
> >
> > > The client drift issue is not clearly defined in this paper.
> >
> > As we state in Sec. 3, client drift arises because by taking multiple local steps in FedAvg, we start overfitting to the individual client’s datasets and not to the overall distribution. This results in FedAvg converging to an incorrect optimum. This observation is not new to our work and was empirically observed by Zhao et al [2018], formalized in Karimireddy et al. [2020], and further refined by [Charles and Konečný 2020]. We defer to these papers for a formal characterization
> >
> > > Definition of $x^\text{out}$?
> >
> > $x^\text{out}$ is randomly chosen from the client updates using a probability distribution defined in the Appendix (Thms II, III, and IV). We didn’t specify it in Thm. I to keep it simple since its definition changes in the PL-strongly convex case vs. the rest.
> >
> > > Spike in MimeSGDm in Fig 2.
> >
> > The spike seen is the characteristic *cyclic* behavior of momentum on strong convex problems (cf. Su et al. [2014]).

---

### Official Review · AnonReviewer1 · 2020-10-28
**I believe this paper adds decent contributions to the ICLR community. The presentation and the structure of the paper can be improved.**

**Rating:** 6
**Confidence:** 2

**Review:**

Summary of the paper:
This paper introduces the MIME framework which can adopt standard centralized SGD methods in federated learning environments and this framework handles the well known "client drift" problem. It is done by cleverly using server statistics such as momentum in local devices. Ample empirical evaluations are provided in order to validate the theoretical claims.

Quality:
Although I did not check the proofs thoroughly, the presented theoretical results looks natural and believable.  Ample experiments are provided to validate the claims. Clarity of the paper is good in general but there are places that can be improved. The quality and the presentation of the paper can be improved significantly.
- Most of the remarks under theorem 1 about the table 1 are redundant. The same remarks are given before the table and sometimes multiple times. For example, the claims about the case where $\delta << L $ and claims about SCAFFOLD. These can be clearly stated in contributions.
- In theorem 1, $\mathbb{E}[||\nabla  f (x^{out})||^2] \le \epsilon$, expectation over what? I believe it is $i$?
- In assumptions A1, A2, the bound on gradient is defined over the expectation of local datasets $i$ but the Hessian bound is for all $i$. Is there any specific reason for this? Or is this a standard assumption?
- Typos: Last line in the first paragraph of Related work, there is an extra "?P:". Page 3 before equation 1, "vour" -> "our".  In equation 5, nothing written after "where".


Originality and significance:
I believe this paper adds decent contributions toward understanding of federated learning and and tackles the "client drift" problem in a simple way.


Pros:
- Nice to have a flexible framework that can adopt standard centralized methods with similar guarantees.
- Extensive experiments validates the provided theoretical claims.

Cons:
- This work is incremental on the paradigm of using momentum in order to improve distributed SGD methods.

Other comments:
The paper titled "On the Linear Speedup Analysis of Communication Efficient Momentum SGD for Distributed Non-Convex Optimization" seems to be discussing a similar approach that uses momentum updates in locally. Can the authors distinguish between these two results?


===========================================================================================================

Added after author response:

--------------------------------------------------------------------------------------------

I have read the author responses and other reviews. I believe authors have a sufficiently addressed my concerns regarding the quality of the paper. I understand the improvement in using momentum in this way and I think it is decent contribution. However, considering this improvement and the concerns raised by other reviewers, I maintain my score.

---

> ### Author Response · Authors · 2020-11-14
> **Improved theory represents a significant step in distributed/federated methods.**
>
> We thank the reviewer for their appreciation of our empirical work, as well as the detailed comments and the useful tips on presentation. We address some specific points below. We hope that this addresses all concerns of the reviewer, and if so request them to reevaluate the significance of our contributions.
>
>  > What is the expectation over In Theorem 1?
>
> The expectation in $E \| \nabla f(x^{\text{out}})\|^2 \leq \epsilon$ is taken both over the sampling of the clients during the running of the algorithm, the sampling of the mini-batches in local updates, and the choice of $x^{\text{out}}$ (which is chosen randomly from the client iterates $y_i$ as described in the Appendix).
>
> > Why is the Hessian bound required almost surely?
>
> This is an excellent question. Hessian distance is indeed a standard notion in distributed optimization and is known to tightly control the rounds of communication required (see [Arjevani & Shamir 2015]). Intuitively, one can think of A1 as a *variance bound* defined over stochastic gradients and A2 as a *smoothness bound* and is defined over the Hessians. We updated Appendix D.1 with more detailed discussions of these assumptions.
>
> >  Incremental on the paradigm of using momentum in order to improve distributed SGD
>
> We are the first to prove theoretically that momentum can improve the rate of convergence in federated and distributed learning and mitigate non-iidness. E.g. while [Yu et al. 2019] show that SGDm converges in the distributed setting, the rates obtained are slower than simply running baseline distributed SGD and imply that we use a very small momentum (which contradicts experimental findings). In contrast, our analyses of momentum yields better rates and implies that we should actually use a very high momentum (in a manner described by MimeMVR) to achieve optimal rates.
>
> This improvement in the rate of convergence is not merely a theoretical curiosity, but represents a major improvement in the design of distributed/federated methods for deep learning. Our generic framework Mime was reverse-built to ensure that when combined with momentum, it yields the optimal rates. Then, experimentally we validated these aspects of Mime design. Such an analysis-first approach is quite rare in deep learning (let alone distributed/federated learning) and is only possible if the analysis closely reflects reality. We hope that the proof techniques developed in this paper can be adapted to derive principled algorithms in a variety of other distributed and federated settings.
>
> References
> - Arjevani and Shamir.  Communication complexity of distributed convex learning and optimization. NeurIPS 2015.
> - Arjevani, et al. Lower bounds for non-convex stochastic optimization. 2019.
> - Yu et al.  On the linear speedup analysis of communication efficient mo-mentum sgd for distributed non-convex optimization. 2019.

---

### Official Review · AnonReviewer4 · 2020-10-29
**Very interesting algorithm utilizing global statistics.**

**Rating:** 4
**Confidence:** 4

**Review:**

In the paper, authors proposed to utilize the global statistics in local client updates to mitigate the client drift problem. The algorithm mine and minelite are very interesting and look reasonable to me. They also prove the convergence rate of the proposed algorithm and evaluate its empirical performances on synthetic and real federated learning simulations.

The following are my concerns:
1) At page 2, analysis of fedavg, the last sentence "technically, it was a slightly modified FEDAVG version? P: FEDAVG as it is understood encompasses methods with server lr." looks very strange to me.
2) typo in the sentence of eq (1) "vour".
3) A more straightforward compared algorithm is fedavg using sgdm on clients.
4) I am confused about the performance of fedsgdm and fedadam in the paper. According to "Adaptive federated optimization" paper, the accuracy of fedsgdm and fedadam achieve about 85%. However, in the paper, the results are far from that.
5) The same question also applies to cifar100 experiments. Besides, it looks like that mimeadam stops converging after 600 rounds, however, fedadam will continue converging to about 50% according to the  "Adaptive federated optimization" paper.

====
After rebuttal:

I read all reviews and rebuttals, especially the argument about Lemma 7. I agree with reviewer 2 that the Lemma 7 is not correct. The assumption of $\mu\eta < \frac{1}{42k}$ contradicts with the requirement $\mu\eta > \frac{1}{k}$ of term 4.  This error needs to be resolved before acceptance.

---

> ### Author Response · Authors · 2020-11-14
> **Differences from Adaptive federated optimization due to amount of hyper-parameter tuning**
>
> We thank the reviewer for their enthusiastic response and the careful reading of the paper. The main concern seems to be comparison with the experimental results of "Adaptive federated optimization" [Reddi et al. 2020], which we explain below.
>
> As we expand upon in Sections A.2 and A.4, there are two main differences between Reddi et al. and Fig. 3: i) we use a 2 layer fully connected network instead of a conv network, and ii) we only tune one parameter--the learning rate and keep all others (such as $\epsilon$ and server learning rate) fixed. For these reasons FedSGDm and FedAdam perform worse in our experiments. The performance of FedAdam on Cifar also starts to decrease after 1k rounds.
> We believe that our *low tuning* setting (7 hyper-param configs) where only the learning rate is tuned is more realistic in federated learning since even running a single 1k round experiment takes about 2 days in the real world (see Sec. A.2). We also replicate the *high tuning* setting (108 hyper-param configs) used in [Reddi et al. 2020] in Table 3, achieving **86%** with a CNN on EMNIST62. Mime still outperforms FedAvg here, but the gap is reduced.

---

### Decision · Program_Chairs · 2021-01-08
**Final Decision**

**Decision:**

Reject

**Comment:**

The submission introduces a theoretically-justified solution to 'client-drift' in federated learning.  Generally, the reviewers agreed that this could be a strong paper, and several of their minor concerns have been addressed in the rebuttal. Unfortunately, a major issue raised during the discussion was the correctness of Lemma 7. Despite the extra clarifications provided by the authors, Reviewer2 still believes it is incorrect and R4 also sided with them.  As theoretical analysis is the major contribution of this work, we have to reject the submission. I would strongly encourage the authors to fix the issue (or clarify the proof), and resubmit it to one of the upcoming top ML conferences.